# Rational design of spontaneous reactions for protecting porous lithium electrodes in lithium–sulfur batteries

Y.X. Ren[1,2], L. Zeng[1,2,3], H.R. Jiang [1,2], W.Q. Ruan[1,2,4], Q. Chen [1,2,4] & T.S. Zhao [1,2]

A rechargeable lithium anode requires a porous structure for a high capacity, and a stable electrode/electrolyte interface against dendrite formation and polysulfide crossover when used in a lithium-sulfur battery. Here, we design two simple steps of spontaneous reactions for protecting porous lithium electrodes. First, a reaction between molten lithium and sulfur-impregnated carbon nanofiber forms a fibrous network with a lithium shell and a carbon core. Second, we coat the surface of this porous lithium electrode with a composite of lithium bismuth alloys and lithium fluoride through another spontaneous reaction between lithium and bismuth trifluoride, solvated with phosphorous pentasulfide, which also polymerizes with lithium sulfide residual in the electrode to form a solid electrolyte layer. This protected porous lithium electrode enables stable operation of a lithium-sulfur battery with a sulfur loading of 10.2 mg cm$^{-2}$ at 6.0 mA cm$^{-2}$ for 200 cycles.

[1] HKUST Energy Institute, The Hong Kong University of Science and Technology, Clear Water Bay, 999077 Kowloon, Hong Kong SAR, China. [2] Department of Mechanical and Aerospace Engineering, The Hong Kong University of Science and Technology, Clear Water Bay, 999077 Kowloon, Hong Kong SAR, China. [3] HKUST Jockey Club Institute for Advanced Study, The Hong Kong University of Science and Technology, Clear Water Bay, 999077 Kowloon, Hong Kong SAR, China. [4] Department of Chemistry, The Hong Kong University of Science and Technology, Clear Water Bay, 999077 Kowloon, Hong Kong SAR, China. Correspondence and requests for materials should be addressed to Q.C. (email: chenqing@ust.hk) or to T.S.Z. (email: metzhao@ust.hk)

The developments of electric vehicles and grid-scale energy storage have been relying heavily on lithium-ion batteries. Yet the low capacity of intercalation electrodes limits the further increase in the energy density of conventional lithium-ion batteries, which motivates our investigations of new materials[1,2]. There is also an ever-increasing interest in revisiting lithium (Li) anode, as Li has the lowest electrode potential (−3.04 V vs. SHE) and the highest capacity (3860 mAh g$^{-1}$) among all anodes[3]. Pairing the Li anode with a high-capacity sulfur cathode, we may achieve an energy density of 500 Wh kg$^{-1}$, doubling that of current Li-ion batteries. However, structural and chemical irreversibility encumbers its stable cycling. On the one hand, Li metal is reactive, continuously consuming electrolyte, electrolyte additives, and active materials. On the other hand, the volume change of Li is infinite, because of its hostless nature, so a solid electrolyte interphase (SEI) required for suppressing side reactions can be repeatedly breached, resulting in a vicious circle of anode degradation.

The infinite volume change of Li metal over cycling requires a porous scaffold to stabilize the electrode structure and to provide more reaction sites for Li plating/stripping. Typical freestanding scaffolds include layered reduced graphene oxide, porous nickel and copper foams[4–7]. Among them, porous carbonaceous scaffolds are attractive for their small pore size, low cost and light weight, but to load them with Li, the surface of the scaffolds needs to be highly lithiophilic. Even when we improve the lithiophilicity by decorating the carbon surfaces with materials like ZnO, Si, Ag, and Au, the molten Li necessitates a high temperature to lower its viscosity, which aggravates the unwanted reactions between Li and trace amounts of environmental impurities (e.g. N$_2$ and O$_2$) that in turn hinder the infusion[8–12]. In spite of the difficulty of fabrication, when the porous Li scaffold is employed in a Li–S full battery, a protective, ion-conductive surface layer is necessary to block the crossover of polysulfide intermediates. Inorganic solid electrolytes (e.g. Garnet-based) have been employed as the surface protective layer[13]. Though effective for the suppression of side reactions as well as dendrite propagation, the use of inorganic solid electrolyte suffers from the fragility, the high Li$^+$ transport resistance, and the relatively low volumetric energy density[14].

Moreover, inside the porous Li scaffold, liquid electrolyte is usually needed for wetting the Li surface. A robust artificial SEI for smoothing the Li deposits and suppressing the consumption of liquid electrolyte thus would be necessary. The SEI needs to satisfy requirements of high chemical stability, high uniformity, adequate elasticity and high Young's and shear modulus[15–21]. Interfacial engineering is usually applied to achieve the above requirements by coating Li surface with mechanically stiff inorganic compounds (carbon, BN, SiO$_2$, Al$_2$O$_3$, etc.); binary Li–X compounds (Li$_3$N, Li$_2$O, Li$_2$S, LiF, Li$_{4.4}$Sn, Li$_{13}$In$_3$); ternary Li–M–X compounds (Li$_3$PO$_4$ and Li$_3$PS$_4$)[22–31]. Among them, LiF is considered as one of the most promising choices for the suppression of side reactions and dendrite growth owing to its high stability and high mechanical strength (Young's modulus: 65 GPa; shear modulus: 55 GPa)[32–35]. By treating the porous Li scaffold in the fluorine-rich gas atmosphere, the LiF-based artificial SEI could be formed across the electrode[36,37]. Such a vapor deposition method, however, might lead to safety concern during scaled-up production. Also, the issues, such as the loose binding between Li and LiF, and low ionic conductivity of LiF remain unsolved[38]. Spontaneous reactions at liquid/solid interface are arguably the most promising approach, recent examples of which include the use of InF$_3$ as an electrolyte additive and CuF$_2$ as the precursor for in-situ and ex-situ SEI formation, respectively[39,40]. However, neither work could generate a sufficiently robust and conductive layer that is also immune to the attack by polysulfides; InF$_3$ is too low in solubility; and CuF$_2$ produces Cu which is unstable towards polysulfides[41].

With these considerations in mind, we design two steps of spontaneous reactions to achieve a porous Li electrode protected by functional composites. In the first step, we infuse molten Li into a sulfur/carbon nanofiber scaffold driven by the sparkling exothermic reaction between Li and the pre-impregnated sulfur to form a porous Li composite. In the next step, the porous Li composite reacts with a metal fluoride complex (BiF$_3$–P$_2$S$_5$) to spontaneously form Li$_3$Bi alloy and LiF that are tightly anchored to the Li surface. P$_2$S$_5$ also polymerizes with residual Li$_2$S on the porous electrode to form an ion conductive amorphous Li$_2$S–P$_2$S$_5$ solid electrolyte layer. The spontaneously achieved multiscale protection enables the stable operation of Li–S full batteries with a rather high sulfur loading (10.2 mg cm$^{-2}$) at 6.0 mA cm$^{-2}$ for over 200 cycles.

## Results

**Fabricating a porous Li electrode**. The porous scaffold for Li we chose is a carbon nanofiber matrix (CNF) of a high specific surface area and a robust interwoven structure (Supplementary Fig. 1). Electrodeposition of Li led to a non-uniform distribution of Li, mostly at the separator side due to its high local Li$^+$ concentration. Such an electrode could not achieve stable cycling (Supplementary Fig. 2). Alternatively, direct thermal infiltration of molten Li encountered the surface lithiophobicity issue, completely preventing the Li infusion[12].

To tackle this issue, we applied the reactive wetting strategy to drive the Li infusion with a reaction between Li and sulfur. We chose sulfur out of a variety of materials after comparing their melting points and Gibbs free energies of complete lithiation reactions (Supplementary Fig. 3). The low melting point of sulfur and its high affinity with carbon allows for its effective filling into a porous scaffold at a low temperature. Besides, its highly exothermic reaction with Li can provide much heat to reduce the viscosity of molten Li and a chemical potential gradient that drives the wetting of Li on the carbon surface to overcome friction and surface tension (Fig. 1a)[42]. Therefore, we first impregnated the CNF with sulfur to form a composite called herein as S–CNF as shown in the scanning electron microscopy (SEM) image (Fig. 1b), which led to a uniform sulfur film on all carbon fiber surface. When we further infiltrated the as-prepared S–CNF scaffold with molten Li, we observed that Li rapidly reacted and wetted the carbon fiber scaffold as shown in Fig. 1c. The infusion of molten Li into a 50-μm-thick CNF scaffold finished within 120 s to form the Li–S/CNF composite. Even with a volume expansion of 70% in such a small duration, the interwoven structure of CNF well accommodated the infused Li (Supplementary Fig. 4). The void space was uniformly filled, as seen in the cross-sectional view (Fig. 1d). Interestingly, after the gravitational condensation, we can create a Li$_2$S-rich surface and a Li-rich surface as shown in Supplementary Fig. 5a, b. This is because the Li$_2$S with a higher density (1.66 g cm$^{-3}$) has poor affinity with both carbon and Li, so it can gradually condense on the electrode's bottom surface to form a dense layer as shown in Supplementary Fig. 5c, d. Differently, lithiated CNF can trap molten Li with its good lithiophilicity[43,44]. Therefore, on the Li-rich surface, we observe the carbon nanofiber was coated with a Li shell as shown in Supplementary Fig. 5e, f. The detailed gravimetric and volumetric fractions of these electrode components are shown in Supplementary Fig. 6 (calculation details can be found in Supplementary Fig. 7. Porosity could be estimated around 7.1 % for the Li/S–CNF electrode, which would be filled with the liquid electrolyte. The chosen initial sulfur loading was 2.5 mg cm$^{-2}$, which guaranteed the uniform Li filling, and an adequate scaffold expansion ratio (Supplementary Figs. 7, 8 and Note 1).

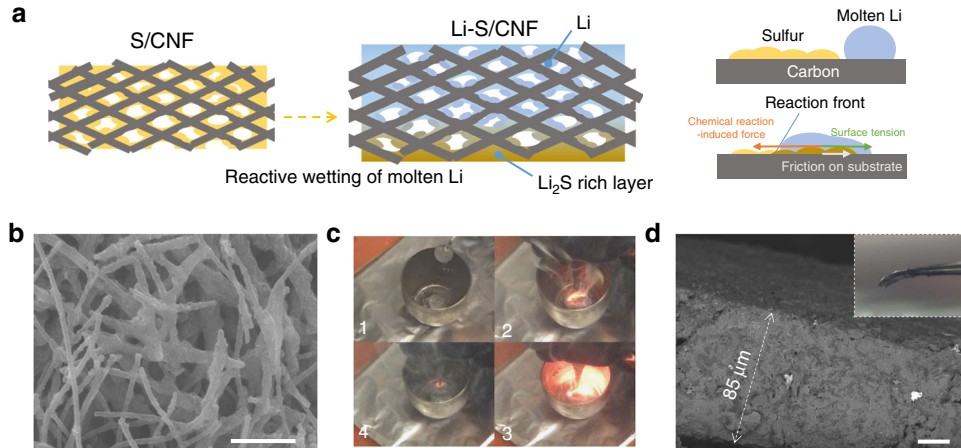

**Fig. 1** The fabrication of Li–S/CNF. **a** Cartoons illustrating the fabrication process. **b** SEM of the surface of the pristine CNF; **c** snapshots taken during the fabrication, showing the sparkling reaction between molten lithium and the sulfur-impregnated CNF; **d** cross-sectional SEM of the Li/S–CNF. Scale bar for **b**, **d** is 5 and 20 μm, respectively

**Protective coating with solubilized metal fluoride**. Next, we need to build an artificial SEI inside the porous Li scaffold. In this step, we employed another spontaneous reaction between Li and $BiF_3$. We chose $BiF_3$ as its reaction with Li can create a composite coating of $Li_3Bi$ and LiF. The Young's modulus of $Li_3Bi$ (22.20 GPa) is much higher than Li metal (1 GPa), which can suppress dendrite formation and strongly interconnect LiF on the Li surface[45]. In addition, $Li_3Bi$ is reported to possess a high concentration of interstitials, which may facilitate the $Li^+$ transport within and through the artificial SEI[38,46]. Moreover, $Li_3Bi$ is electrochemically inert given its high delithiation voltage of ~1.1 V vs Li/Li⁺ (Supplementary Fig. 9 and Note 2). As a step further, we corroborate our postulation on the benefit of $Li_3Bi$ and LiF with first-principle density functional theory-based (DFT) calculations. The results show that Li adatom is weakly adsorbed on the substrate of $Li_3Bi$, providing the lowest surface diffusion energy barriers for Li adatom (0.17 eV), followed by LiF (0.36 eV), and then Li metal (0.39 eV) (Fig. 2a, b, Supplementary Figs. 10 and 11). We thus expect that a surface composite layer of LiF and $Li_3Bi$ can result in facile Li plating and smooth Li deposits.

The benefits of this composite layer, however, should not dwarf the difficulty of reacting a compound of $BiF_3$ with the Li/S–CNF composite uniformly, due to the high melting point of $BiF_3$ (649 °C) and its insolubility in most solvents. Our strategy is solubilizing $BiF_3$ with $P_2S_5$ as a complexing agent. $P_2S_5$ has been previously reported to complex with $Li_2S$ to form solid electrolyte precursor, because sulfide anions can initiate nucleophilic attack on the electrophilic phosphorous in $P_2S_5$, causing ionic polymerization[27]. Here we show that complexation occurs between $BiF_3$ and $P_2S_5$. The lone pair electron of Bi can initiate nucleophilic attack toward P to form two possible products as shown in Fig. 2c. We measured ³¹P nuclear magnetic resonance spectroscopy (NMR) spectra of the supernatants of mixtures of 50 mM $BiF_3$ with various amounts of $P_2S_5$ in DME (Fig. 2d) to verify our hypothesis. Comparing the NMR spectra, we can deduce that two peaks at 105.2 and 97.8 ppm represent the $BiF_3$–$P_2S_5$ complex with one phosphorous reacted, which will take a higher portion in the soluble product when the molar ratio of $BiF_3$: $P_2S_5$ decreases. And the peak at 83.1 ppm represents the complex with both phosphorous atoms reacted. This complex likely maintains the symmetrical adamantane-like cage structure, therefore showing the single peak in NMR. While $BiF_3$ or $P_2S_5$ alone has intrinsically negligible solubility, for the solution with a $BiF_3$: $P_2S_5$ ratio of 1:1, the solubility of $Bi^{3+}$ is 570 ppm, as

determined by the inductively coupled plasma optical emission spectrometry (ICP-OES)[47]. The as-prepared mixture will then be used to react with Li to form a composite artificial SEI comprising of LiF and $Li_3Bi$ (Fig. 2e). In spite of using ether-based solvent like DME, similar complexation and solvation behavior can also be attained in ester (e.g. ethyl propionate) and nitrile (e.g. acetonitrile) based solvents as shown in Supplementary Fig. 12.

We first applied this solution-based coating strategy to the planar Li foil. The reduction of $BiF_3$ on the Li surface resulted in the formation of a uniform, conformal black layer around 5 μm in thickness as observed from the cross-sectional SEM images (Supplementary Fig. 13). This protected Li foil will be referred to herein as metal fluoride complex protected Li electrode (MFC-Li for short). The X-ray diffraction (XRD) results indicate the existence of $Li_3Bi$ and LiF phases (Supplementary Fig. 14), which can be further confirmed by the evolution of the X-ray photoelectron spectroscopy (XPS) spectra of Bi 4f and F 1 s after lithiation (Fig. 3a and Supplementary Fig. 15). No signals of $BiF_3$ or Li metal can be found in XRD or XPS, confirming the complete reaction. We also analyzed quantitatively the composition of the coating along the depth direction via XPS spectroscopy by sputtering its surface by Ar gas for different durations. The results indicate a uniform distribution of LiF and $Li_3Bi$ alloy (Fig. 3b), with small contents of P and S (Fig. 3c), which is likely the residual $P_2S_5$ and the reaction products between Li and $P_2S_5$ (grey fitting line in Fig. 3a). The $Li_3Bi$/LiF layer indeed provides a high ionic conductivity, estimated to be $6.9 \times 10^{-4}$ S cm⁻¹, with the layer thickness measured with cross-sectional SEM and time of flight-secondary ion mass spectroscopy (TOF-SIMS) and the resistance determined by the use of electrochemical impedance spectroscopy (EIS) (Supplementary Fig. 13 and Note 3). This value is much higher than that of defect-free LiF (~$10^{-12}$ S cm⁻¹)[36].

The microscopic structure of the coating is shown in Fig. 3d. Although we could not ensure the integrity of the coating layer after stripping it off the sample for transmission electron microscopy (TEM) analysis, with the characterization we can confirm the uniform distribution of species (Fig. 3e, f). At a higher magnification, we can further identify the interconnected, nanometer-wide domains of LiF and $Li_3Bi$ (Supplementary Fig. 16). The abundant $Li_3Bi$/LiF interfaces might accelerate the $Li^+$ transport due to the space-charge effect, explaining the high ionic conductivity achieved[36]. One potential concern of the effectiveness of this layer is the possibility of Li plating on its top, which is however unlikely to happen owing to the relatively high

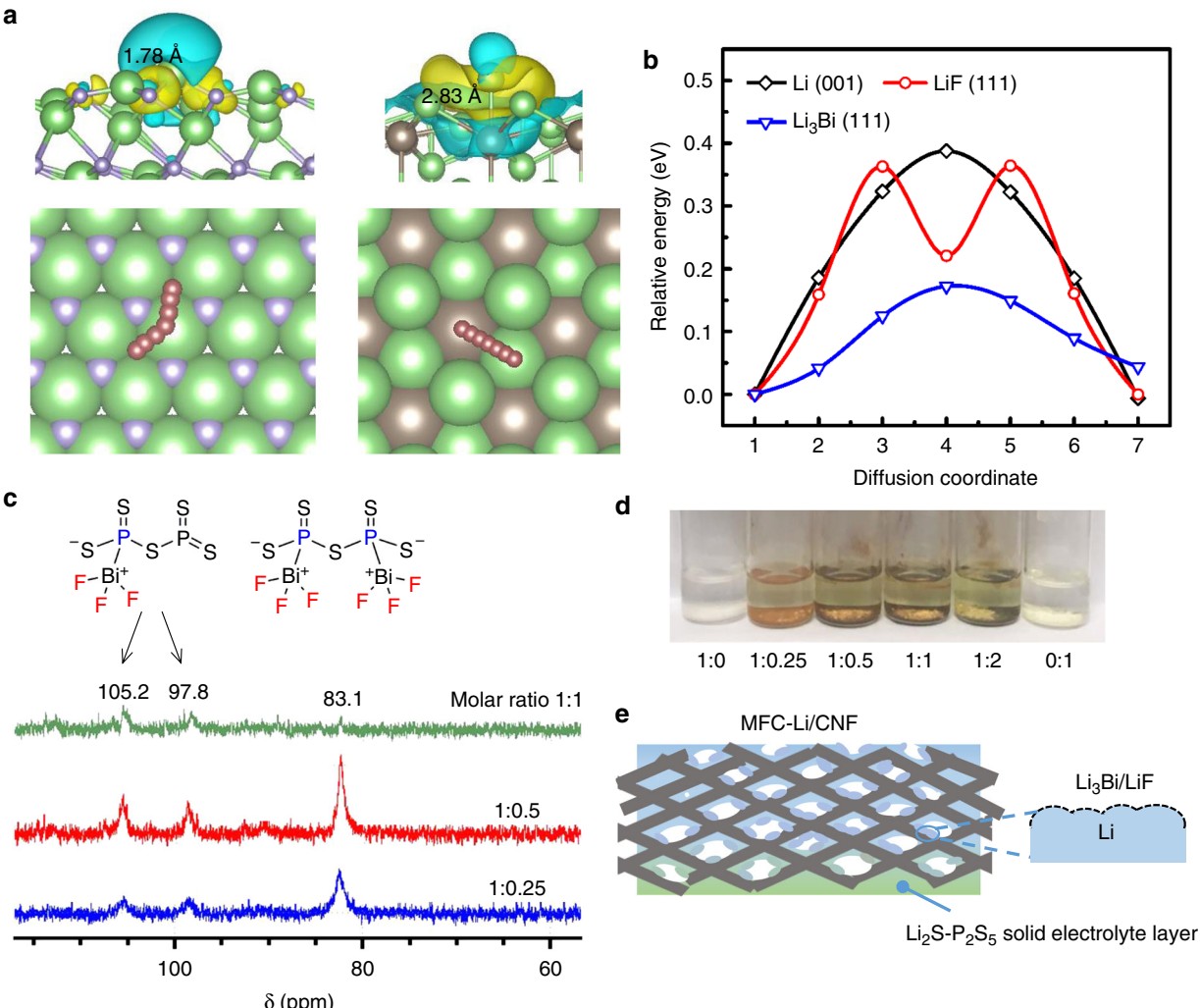

**Fig. 2** Coating Li by solubilized $BiF_3$. **a** Charge density difference plots showing the adsorption of a Li adatom on the substrate of LiF (111) (left) and $Li_3Bi$ (111) (right) (top row); the diffusion pathways of Li adatoms on the corresponding surfaces (bottom row). The green, purple and brown balls represent lithium, fluorine and bismuth atoms, respectively. **b** Comparison of diffusion energy barriers on Li (001), LiF (111) and $Li_3Bi$ (111) facets. **c** The proposed reaction products of the complexation between $BiF_3$ and $P_2S_5$, and $^{31}P$ NMR solution spectra of the supernatants of $BiF_3$ and $P_2S_5$ mixture at different molar ratios ($BiF_3$:$P_2S_5$) as labeled. **d** Photography of the mixtures of $BiF_3$-$P_2S_5$ with different molar ratios ($BiF_3$:$P_2S_5$) in DME. **e** A cartoon showing the coating strategy to derive MFC-Li/CNF

electronic resistance as we observed in the earlier EIS test (Supplementary Fig. 13d). We confirmed the postulated plating location (beneath the protective coating) by employing this coated Li foil (MFC-Li) in a symmetric cell, at a moderate deposition current density (1.0 mA cm$^{-2}$, 1.0 mAh cm$^{-2}$) as shown in Supplementary Fig. 17a, b, we achieved stable cycling for 300 cycles. Over cycling, the electrode retained a rather thin layer of Li beneath the protective layer and a smooth surface morphology (Supplementary Figs. 18 and 19). However, at a higher rate (e.g. 2.5 mA cm$^{-2}$, 1.0 mAh cm$^{-2}$) the overpotential promoted to twice of its initial values after 300 cycles (Supplementary Fig. 20), indicating the necessity of the porous scaffold.

**A protected porous Li electrode**. We then applied the coating strategy to the Li/S–CNF composite to form a metal fluoride complex coated Li/CNF composite electrode, referred to herein as MFC-Li/CNF, by soaking it in the $BiF_3$–$P_2S_5$ mixture. In this step, due to the reaction between Li and solvated $BiF_3$, there will be a spontaneous formation of $Li_3Bi$ and LiF composite, which functions as the artificial SEI on the Li-coated fibers. Meanwhile, $P_2S_5$

in the solution polymerized with the residual $Li_2S$ on the scaffold to form the solid electrolyte precursor. Upon subsequent annealing at 160 °C and mechanical pressing, a layer of light yellow solid electrolyte evolved to cover the Li scaffold surface (Fig. 4a, b), exhibiting a uniform surface morphology. No characteristic peak of this solid electrolyte can be seen in XRD likely due to its amorphous structure (Fig. 4c). From the Raman spectra for the reaction precursor, intermediates and reaction products, the species formed from the polymerization of $P_2S_5$ initiated by $Li_2S$ were identified (sample #2 in Fig. 4d), showing peaks at 218/476, and 248/437 cm$^{-1}$, together with the peaks of $P_2S_6^{4-}$, $P_2S_7^{4-}$ and $PS_4^{3-}$ from 380 to 420 cm$^{-1}$ (Fig. 4d)[48,49]. Further sintering combines these unstable intermediates into a stable phase for the final product, showing a single peak representing $Li_3PS_4$ at 418 cm$^{-1}$[50]. The formation of $Li_3PS_4$ can be confirmed with the S 2p spectrum (Fig. 4e), which exhibits S $2p_{3/2}$ peaks at 159.6 and 161.2 eV corresponding to the P = S and P–S–Li moieties of $Li_3PS_4$ respectively. Besides, the P 2p spectrum at 132.4 eV also corresponds to the formation of $Li_3PS_4$ (Fig. 4f). Comparing these results with the S 2p and P 2p spectra of $P_2S_5$ (Fig. 4e, f), we can confirm the complete reaction between $P_2S_5$

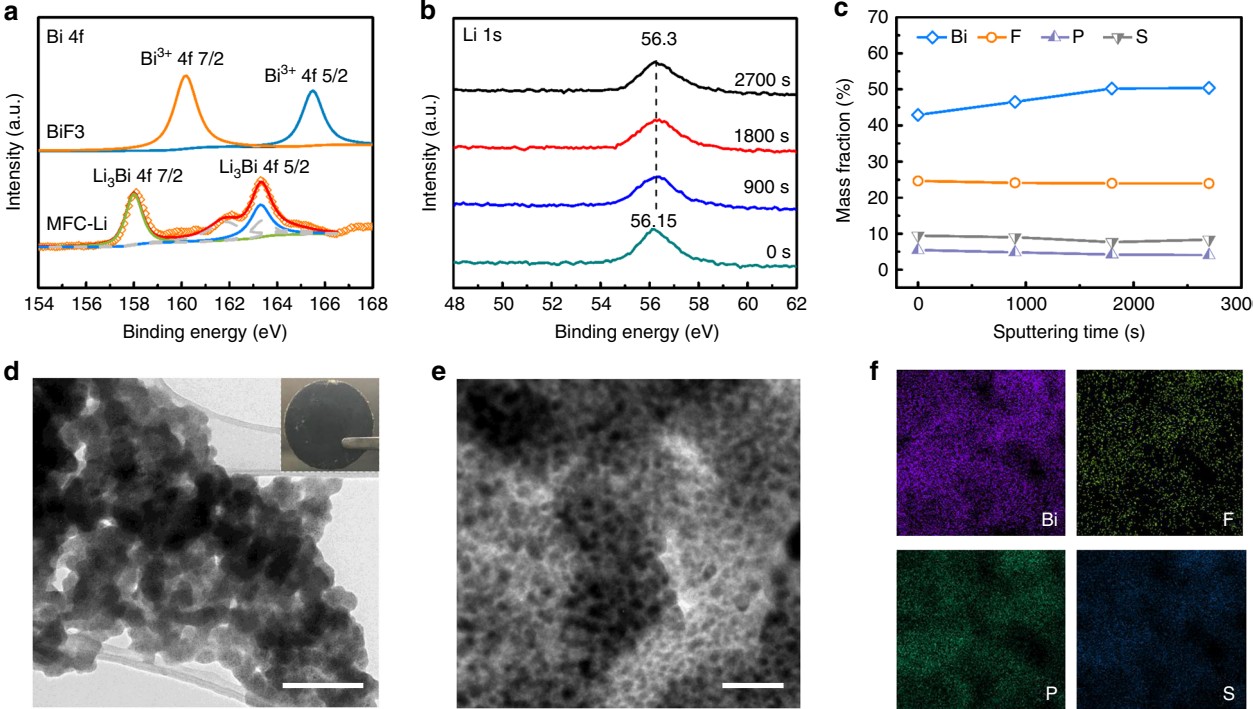

**Fig. 3** The protective layer on a Li foil. **a** XPS spectra of Bi 4 f collected for BiF$_3$ and MFC-Li (after sputtering 1800 s). **b** XPS spectra of Li 1 s collected at different depths. **c** XPS depth profiles of Bi, F, P, and S elements. The sputtering speed was 13 nm min$^{-1}$. **d** TEM image of the protective layer stripped from the MFC-Li electrode; **e**, **f** TEM/EDS analysis of the protective layer. Scale bar for **d** is 200 nm, and inset of **d** is 1 cm; scale bar for **e** is 50 nm

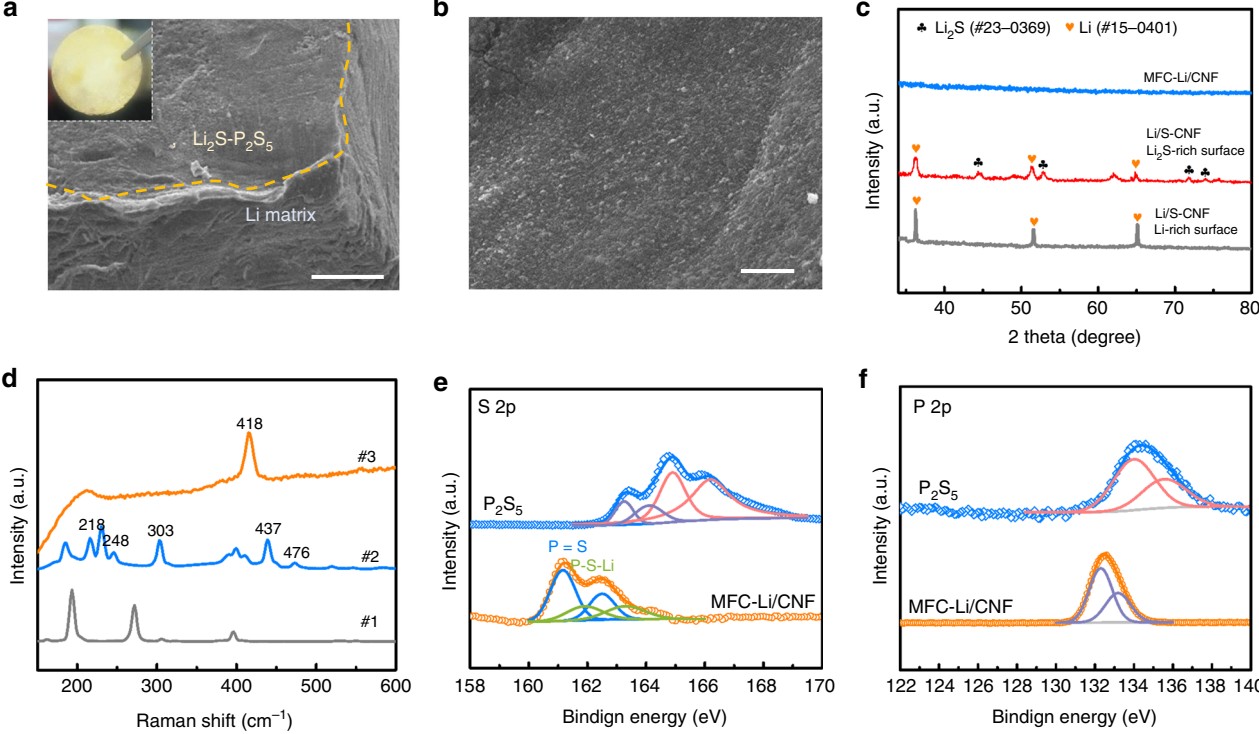

**Fig. 4** Protective coating on the porous Li electrode. **a** SEM image of the edge of MFC-Li/CNF; **b** surface SEM image of MFC-Li/CNF; **c** XRD of the MFC-Li/CNF surface (the same as **b**); Li$_2$S-rich and Li-rich surfaces of the Li/S–CNF. The Li$_2$S-rich surface was formed from the gravitational setting of Li$_2$S during the formation process of Li–S/CNF composite. And the solid electrolyte layer also evolved from the Li$_2$S-rich surface. **d** Raman spectra for P$_2$S$_5$ (#1) and the coated Li/S–CNF electrode after drying (50 °C, #2) and after heat treatment (160 °C, #3). **e**, **f** XPS spectra of S 2p (**e**) and P 2p (**f**) for P$_2$S$_5$ and MFC-Li/CNF. Scale bar for **a** is 100 μm, for **b** is 50 μm

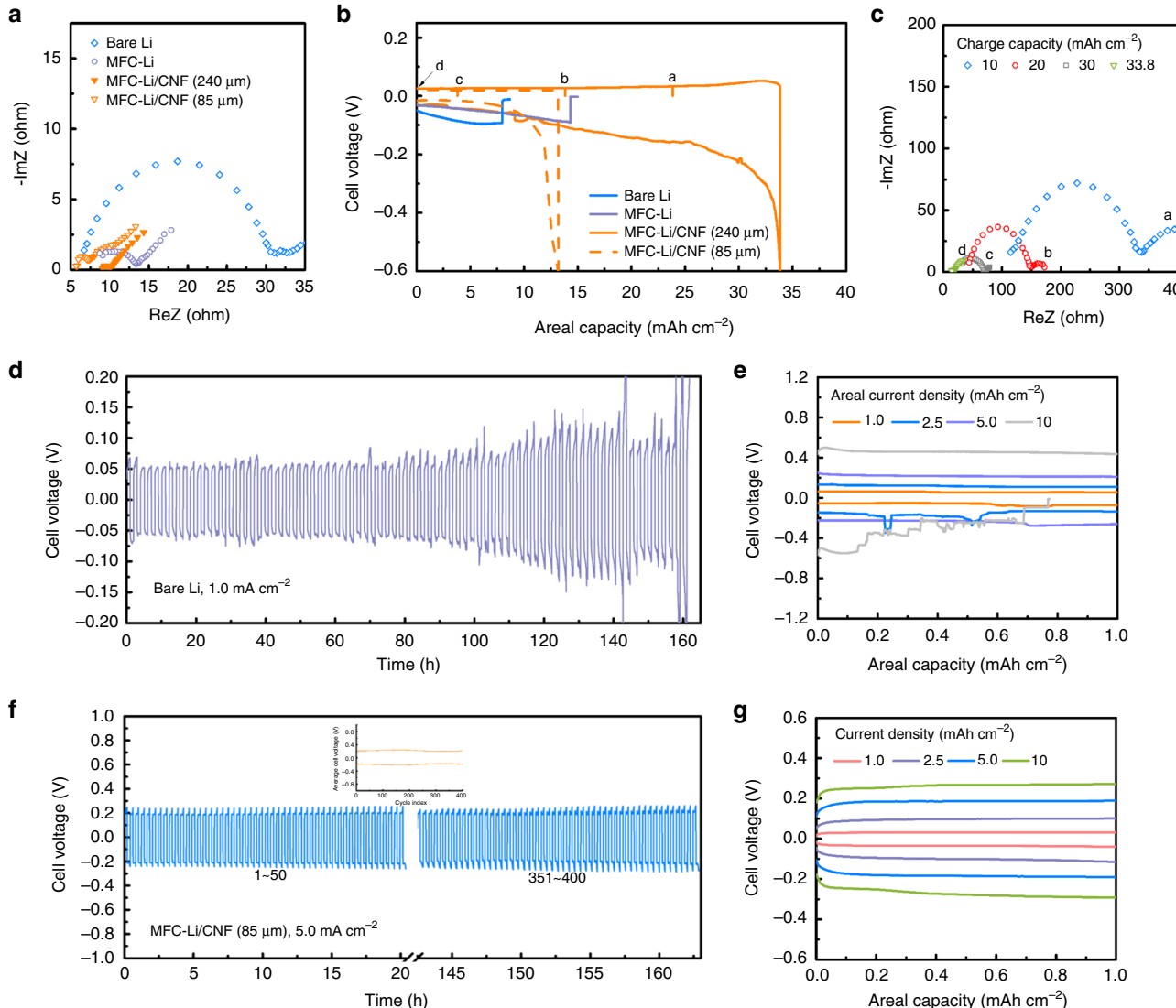

**Fig. 5** Symmetric cell tests with MFC-Li/CNF electrodes. **a** EIS measurement results of symmetric cells assembled with different Li electrodes. **b** Long-term discharging profiles before short circuit (0.5 mA cm$^{-2}$) for the symmetric cell with different Li electrodes; **c** EIS measurement results of the MFC-Li/CNF electrode with different charge capacities. **d** Representative cycling voltage profiles at 1.0 mAh cm$^{-2}$ at 1.0 mA cm$^{-2}$ for the symmetric cell assembled with the bare Li metal (79 cycles); **e** voltage profiles at different current densities. **f** Representative cycling voltage profiles at 1.0 mAh cm$^{-2}$ at 5.0 mA cm$^{-2}$ for the symmetric cell assembled with the MFC-Li/CNF electrode, the inset shows the average cell voltage versus the cycle index; **g** voltage profiles at different current densities

and Li$_2$S on the electrode surface[51]. The formation of the solid electrolyte layer can also be seen in the EIS profile and the polarization test. By applying the coating to a thick Li–S/CNF (240 μm) electrode, the low-frequency real part of the impedance (ReZ) decreased from 12 Ω (Supplementary Fig. 21a) to 9.0 Ω (Fig. 5a), indicating that the formation of a continuous solid electrolyte layer can indeed promote the ion transport. This value decreased to 6.0 Ω if the coating was applied to a thinner porous electrode (85 μm). In Supplementary Fig. 22, the polarization curves further reveal that, for the 85-μm-thick MFC-Li/CNF, because of the formation of a more continuous surface solid electrolyte layer, the transport of electron across the electrode surface can be well insulated, leading to an overall areal resistance of 14.9 Ω cm$^{-2}$, markedly higher than that value of Li/S–CNF. The benefit of this layer of Li$_2$S–P$_2$S$_5$ based solid electrolyte is to prevent the dissolved polysulfides from either corroding or passivating the porous Li electrode, and prevent dendrite penetration with its high stiffness, as we will see later in the battery tests.

**Battery performance**. The capacity, reversibility and stability of the MFC-Li/CNF were further evaluated in the symmetric cell. We began with an 85 μm-thick MFC-Li/CNF electrode. By discharging the electrode at 0.5 mA cm$^{-2}$ to the cut off voltage (−0.6 V) or cell shorting (Fig. 5b), we achieved an areal capacity of 13.2 mAh cm$^{-2}$ (1517 mAh g$^{-1}$ based on the overall electrode weight), with the utilized Li taking up 75.3% volume of the electrode. Similar levels of specific capacity (1352 mAh g$^{-1}$) and volume fraction of utilized Li (68.3 %) were achieved for a thicker electrode (240 μm in thickness). We further show that at such a high areal discharge capacity (33.8 mAh cm$^{-2}$), the counter electrode (also a MFC-Li/CNF) suffered from a loss of porosity, leading to the rise of impedances[52]. During charging, we observed that the impedance spectrum returned to its initial value, confirming the electrode's capability to withstand large volume change (Fig. 5c). In comparison, symmetric cells with neither Li–S/CNF (240 μm in thickness, Supplementary Fig. 21b) nor planar Li electrodes could achieve such a high areal capacity due

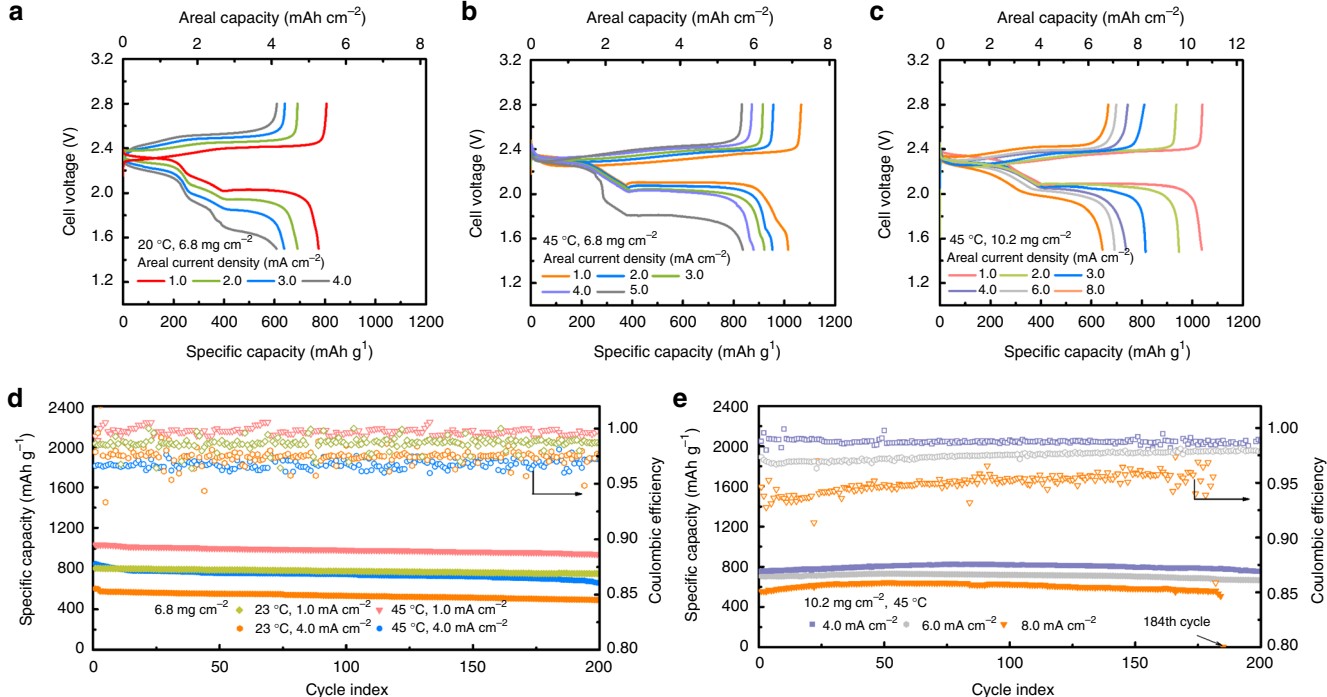

**Fig. 6** Performance of a Li–S battery employing the MFC-Li/CNF electrode. **a**, **b** Voltage profiles of Li–S batteries with a sulfur loading of 6.8 mg cm$^{-2}$ at 23 °C (**a**); and at 45 °C (**b**). **c** Voltage profiles of Li–S batteries with a sulfur loading of 10.2 mg cm$^{-2}$ at 45 °C. **d** Cycling performance with a sulfur loading of 6.8 mg cm$^{-2}$ at 1.0 or 4.0 mA cm$^{-2}$, at 23 °C or 45 °C. **e** Cycling performance with a sulfur loading of 10.2 mg cm$^{-2}$ at 4.0, 6.0 or 8.0 mA cm$^{-2}$ at 45 °C

to the unregulated Li plating. Furthermore, we carried out the long-term cycling test to evaluate the stability of the Li/electrolyte interphase. The planar Li electrode could not be stably cycled at 1.0 mA cm$^{-2}$ within 160 h and exhibited the high-overpotential, fluctuating voltage profiles at a higher current density (Fig. 5d, e). Such a phenomenon was ascribed to the significant heterogeneity on the Li/electrolyte interphase after cycling (Supplementary Fig. 18)[53]. Enlarging the electrode surface area and building robust SEI across Li and liquid electrolyte could smoothen the Li deposits. As a result, the symmetric cell with the MFC-Li/CNF electrode (85 μm) could be cycled for more than 400 times even at 5.0 mA cm$^{-2}$ (Fig. 5f), and exhibited smaller overpotentials from 1.0 to 10 mA cm$^{-2}$ (Fig. 5g).

For the Li–S full battery, we applied the 85-μm-thick MFC-Li/ CNF anode to pair with a sulfur/carbon cloth cathode of a loading of 6.8 mg$_{sulfur}$ cm$^{-2}$ as shown in Fig. 6a, b, d. At 23 °C, the discharge capacity achieved at a cathode current density of 4.0 mA cm$^{-2}$ (anode current density 3.4 mA cm$^{-2}$) was 4.14 mAh cm$^{-2}$. If operated at 45 °C, this value could go up by 44% to 5.96 mAh cm$^{-2}$ (Fig. 6a, b), owing to the enhancements in both ion mobility and reaction kinetics. For long-term cycling (200 cycles), the battery also presented stable performance at both 23 and 45 °C (Fig. 6d). At 1.0 mA cm$^{-2}$ the battery achieved a specific capacity of 942 mAh g$^{-1}$ after 200 cycles (112 days), with a capacity retention of 90.7% and practically utilizing more than 41% Li stored in the anode. At 4.0 mA cm$^{-2}$, the achievable capacity was still as high as 662 mAh g$^{-1}$ after 200 cycles. As comparisons, even with performance enhancement, neither MFC-Li nor Li/S–CNF could deliver similar performance due to a limited surface area or the lack of robust protective layer (Supplementary Figs. 23 and 24). Further, post-mortem structural/compositional characterizations were conducted for the cycled MFC-Li/CNF anode. There was either electrode-level deformation or structural change after high rate cycling (4.0 mA cm$^{-2}$) as shown in Supplementary Fig. 25. From the XPS depth profiling, we can further observe that both the compositions of

Li$_2$S–P$_2$S$_5$ based solid electrolyte layer and Li$_3$Bi/LiF-based artificial SEI retained stable.

To assess the feasibility of our strategy in a more practical cell, we further tested a battery with a 50% higher sulfur loading (10.2 mg cm$^{-2}$) at higher current densities, shown in Fig. 6c, e. Under these conditions, the concentration of Li$_2$S$_8$ intermediate can rise to ~0.98 M in the cathode region during discharge, whose shuttle effect would lead to significant side reactions if there was no adequate anode protection, and the Li anode stress would seriously deteriorate a planar Li anode as shown previously by Jiao et al.[54] At 45 °C, our battery delivered discharge capacities of 7.53 mAh cm$^{-2}$ and 7.05 mAh cm$^{-2}$ at 4.0 mA cm$^{-2}$ and 6.0 mA cm$^{-2}$ respectively. Even at 8.0 mA cm$^{-2}$, it could achieve 6.6 mAh cm$^{-2}$ (Fig. 6c). When we lowered the temperature to 23 °C, the discharge capacity decreased likely due to the slower reaction kinetics in the high-loading electrode (Supplementary Fig. 26), as also shown by Xiao et al.[55] As shown in Fig. 6e, the high-loading battery (10.2 mg cm$^{-2}$) could be cycled at 4.0 and 6.0 mA cm$^{-2}$ at 45 °C for more than 200 cycles stably, with an areal capacity constantly above 7.5 and 6.8 mAh cm$^{-2}$, respectively. If cycled at 8.0 mA cm$^{-2}$, the battery could last for about 180 cycles. Compared with literature, our Li–S battery, either with 6.8 mg cm$^{-2}$ or 10.2 mg cm$^{-2}$ of sulfur loading, stands out in terms of areal capacities, utilization of Li and capacity retention[56–63]. It is particularly remarkable that battery even outperforms those with the addition of LiNO$_3$ (Supplementary Table 1 and Note 4), a popular electrolyte additive that often convolutes benefits from surface protective layers[64]. In addition, given the parasitic gas evolution and the limited benefit in long-term, high-loading cycling, we decide not to explore the use of LiNO$_3$ in this work[65,66].

## Discussion

In summary, we designed two steps of spontaneous reactions to achieve a porous Li electrode protected by functional composites,

which delivered stable cycling performance at 45 °C with an areal capacity above 7.5 mAh cm$^{-2}$ (4.0 mA cm$^{-2}$) and 6.8 mAh cm$^{-2}$ (6.0 mA cm$^{-2}$) in a Li–S battery. Both steps of fabrication can be generalized to other structures or materials. In the first step, the sulfur impregnation can apply to other carbon-based porous frameworks to drive the infiltration of molten Li with the exothermic reaction. In the second step, the complexation strategy to solubilize metal fluoride is by no means limited to $BiF_3$. $SbF_3$ or $InF_3$, can also complex with $P_2S_5$, which can then react with Li to form a composite coating comprising of LiF and Li-alloy (Supplementary Figs. 27–29). It is evident that many parameters in the two steps can be tailored to optimize the anode structure for better performance. In addition, the $BiF_3$-$P_2S_5$ precursor solution and the formed MFC-Li/CNF electrode are both stable in the dry air (Supplementary Fig. 30 and Note 5), which will facilitate cost-effective processing and battery assembly at a practical scale. Therefore, we believe that the facileness and the flexibility of our electrode design strategy holds a high promise in its application in practically rechargeable Li-based batteries.

## Methods

**Sample preparation**. The CNF scaffold was prepared according to the reported method[67]. Briefly, polyacrylonitrile (PAN) fibers were firstly obtained from PAN in dimethylformamide (DMF) solution by electrospinning technique. Stabilization and subsequent carbonization of the electrospun PAN fibers were carried out in a high temperature furnace to form the CNF.

Sulfur was impregnated inside the CNF scaffold (13 mm in diameter) on a hot plate (155 °C). Due to the high affinity of sulfur/carbon, as soon as the CNF and molten sulfur droplet were attached, sulfur droplets could wet the CNF. The Li infusion, coating and heat treatment were all conducted inside the Ar-filled glove box, with $O_2$ and $H_2O$ levels maintained below 1 ppm. When in contact with molten Li (200 °C), the sparking reaction soon occurred, which allowed the infusion of molten Li inside the CNF. After the reactive wetting process, by putting the composite on the heated flat stainless steel plate (200 °C) for 60 min, the gravitational setting of $Li_2S$ formed a $Li_2S$-rich surface layer (Fig. 4c). Beside, excessive molten Li that attached on the carbon surface could be squeezed out of the Li/S–CNF composite. The areal weight of the CNF scaffold was 0.80 mg cm$^{-2}$ and the areal sulfur loading before Li infusion was 2.5 mg cm$^{-2}$. After Li infusion the areal weight increased to 5.5 mg cm$^{-2}$ and thickness increased to 85 μm. With the thicker CNF scaffold (150 μm in thickness), sulfur loading was also tripled, after Li infusion, the areal weight increased to 14.7 mg cm$^{-2}$, and thickness increased to 240 μm.

$BiF_3$ (Sigma-Aldrich, anhydrous, ≥99.9%) together with $P_2S_5$ (Sigma-Aldrich) were dispersed in 1,2-dimethoxyethane (DME) (5 mL, Sigma-Aldrich) by vigorously stirring for 60 min. Prior to use, DME was dehydrated with 3 Å molecular sieves and $BiF_3$ was heated at 120 °C under vacuum both for 48 h. Li foil (13 mm in diameter, 99.9%, Aldrich) was polished one side for coating, while the other side was coated with a removable, dense Kapton (polyimidie) film to prevent the reaction. The Li foil or the Li–S/CNF composite was put into the $BiF_3$–$P_2S_5$ mixture (1:1 molar ratio, 50 mM $BiF_3$ in 5 mL solvent) for 60 min. Afterwards, the coated Li or porous Li composite was rigorously rinsed with pure DME and was heated at 50 °C for 12 h to fully evaporate the solvent. After drying, there was an areal weight increase of 1.8 mg cm$^{-2}$ on the Li foil. For the thinner Li/S–CNF composite (85 μm), the areal weight increase of 3.2 mg cm$^{-2}$ (overall 8.7 mg cm$^{-2}$) was attained after solution coating for 60 min. For the thicker Li/S–CNF composite (240 μm), the areal weight increase of 10.3 mg cm$^{-2}$ (overall 25.0 mg cm$^{-2}$) was attained. After the complete evaporation of solvent at 50 °C, all of the porous Li composites were compressed using a pellet die with a pressure of ~10 metric tons and heat treated at 160 °C to form the MFC-Li/CNF composite electrodes (13 mm in diameter).

**Battery assembly and testing**. CR-2032 coin cell setup was employed for the battery assembly. The typical electrolyte composition is lithium bis(trifluoromethanesulphonyl)imide (LiTFSI, 1 M) in 1:1 v/v 1,3-dioxolane (DOL)/DME.

For assembling Li/Li symmetric cells, bare Li foil, MFC-Li or MFC-Li/CNF with the same diameter of 13 mm was used as the electrode. The two electrodes were saturated with overall 100 μL electrolyte and separated by two pieces of Celgard 2500 separators.

For assembling Li–S batteries, bare Li foil, MFC-Li or MFC-Li/CNF with the same diameter of 13 mm was used as the anode. Carbon cloth (12 mm diameter, ELAT®) was used as the cathode current collector and loaded with different amounts of sulfur (2.8, 6.8 or 10.2 mg cm$^{-2}$). The two electrodes were filled with 100 μL electrolyte, and separated by two pieces of Celgard 2500 separators. The galvanostatic discharge and charge tests were conducted on a battery cycling

system (Neware, CT-4008W) at a constant temperature (23 or 45 °C). Electrochemical impedance spectroscopy (EIS) from 100 kHz to 0.1 Hz with an alternating current amplitude of 5 mV was tested on a potentiostat (Princeton Applied Research, PARSTAT M2273) via the two-electrode setup.

**Characterization**. JSM-6700F field emission SEM instruments were used for micrograph observation at an acceleration voltage of 5.0 kV. TEM images were taken on a high-resolution JEOL 2010F TEM using an accelerating voltage of 200 kV. The protective layer on the planar MFC-Li electrode can be stripped by sonification and dispersed in DME for preparation of the TEM samples.

NMR experiments were conducted with neat liquids on a Bruker Avance 400-MHz instrument at ambient temperature. $^{31}P$ NMR spectra were recorded at 162 MHz. All signals were reported in δ unit with references to the external 85% $H_3PO_4$ for phosphorus chemical shifts. ICP-OES measurements were carried out using the ESCALAB250 version made in company THERMO in U.S.A.

The XRD pattern was analyzed with a Philips high-resolution X-ray diffraction system (model PW 1825) using a Cu Kα source operating at 40 kV with the scan rate of 0.025° s$^{-1}$. To protect the highly reactive Li-related compound from the air, the XRD samples were loaded on a glass slide and covered with Kapton tape in the glove box before the measurements. XPS measurements were conducted on a Physical Electronics PHI5802 instrument using an X-rays magnesium anode (monochromatic Ka X-rays at 1253.6 eV) as the source. For the Raman study, modification of a regular single cell was made. The cathode end cap plate was cut with a hole and sealed with a quartz window. Raman spectra were collected using a spectrometer (Princeton Instruments, Spectrapro 2500i) with a wavelength of 514.5 nm and 50 mW laser power as the excitation source. The TOF-SIMS test was conducted using TOF-SIMS IV (ION-TOF GmbH, Germany) with a 25 keV. The negative secondary ions were induced by the primary ion beam bombardment on the surface of Li electrodes. Depth profiles were obtained by sputtering ion beams of Cs$^+$ (3 keV) on square areas of $150 \times 150$ μm$^2$. The analysis area was $19 \times 19$ μm$^2$. The UV-Vis spectra were collected by SEC2000 UV-visible spectrophotometer (ALS Co., Ltd.). An SCE-C thin layer quartz glass cell with an optical path length of 4.5 mm was used as the holder.

**Modeling**. DFT based first-principles calculations were carried out using ABINIT code adopting the generalized gradient approximation (GGA) of Perdew-Burke-Ernzerhof (PBE) type[68]. The electron-ion interactions were taken into account by Projector-augmented-wave (PAW) potentials[69]. The energy cutoff and the thickness of vacuum layer were set to 24 Ha and 20 Å, respectively. The facets were chosen based on the TEM and XRD characterization ((111) facet for both $Li_3Bi$ and LiF) as shown in Supplementary Fig. 16. The spacing between k-point grids were all set to be <0.05 Å$^{-1}$. The slab models contain four layers, with the bottom two layers fixed during structural optimization. The force convergence criteria for self-consistent-field cycles and structural relaxation were set to $4.0 \times 10^{-5}$ and $6 \times 10^{-4}$ Ha Bohr$^{-1}$.

The adsorption energy ($E_{ads}$) of lithium adatom on slab was calculated by:

$$E_{ads} = E_{Li} + E_{slab} - E_{system}$$

where $E_{Li}$, $E_{slab}$, and $E_{system}$ are the DFT energies of Li atom, slab and adsorption systems, respectively. To calculate the lithium diffusion barriers, the simplified string method was adopted[70].

## Data availability

The data that support the plots within this paper and other finding of this study are available from the corresponding author upon reasonable request.

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

## Acknowledgements
The work described in this paper was fully supported by a grant from the Research Grants Council of the Hong Kong Special Administrative Region, China (Project No. T23-601/17-R).

## Author contributions
T.S.Z. and Q.C. supervised the research; Y.X.R. conceived the experiments; Y.X.R. and H.R.J. designed the numerical simulations; Y.X.R., L.Z., H.R.J. and W.Q.R. analyzed the data; Y.X.R., Q.C. and T.S.Z. wrote the paper and all the authors discussed the paper.

## Additional information

**Competing interests:** The authors declare no competing interests.

