## [Peer Review File · Nature Communications]

Reviewers' comments:

Reviewer #1 (Remarks to the Author):

The work of "Rational design of spontaneous reactions to form a protected porous Li electrode for a stable Li-S battery" by Ren et al. has been reviewed carefully. In the work, the authors developed two steps of Li anode engineering that involved Li reaction with a porous S/C structure and BiF₃, in which the protected Li electrodes show good cycling performance. However, there are several points listed below as reviewer's comments that need to be clarified first. Although the work was characterized well, there are many similar works already published. The reviewer suggests the authors to provide more systematic and quantitative comparison between this work and previous published work to see where this work stands out.

Comments:

1. There are many works done on protecting Li for Li-S chemistry with 3D porous scaffolds, e.g. the work by Lin et al. (Nature Nanotechnology volume 12, pages 194–206 (2017)), and some works on forming protective layers with electro-polymerization e.g. the work by Wang et al., (DOI: 10.1021/acsami.8b07248). Therefore, this work is not with high novelty. The reviewer suggests the author to systematically compare their design and outcome to the already published work to see if this work stands out from previous works.

2. Keep in mind – in Li-S battery, the ratio between electrolyte volume and sulfur loading (E/S) is an important factor (E/S should be low) for it to be useful. The design of porous Li electrode will need more electrolyte to wet the interfaces. Do the authors have estimation on how much porosity and electrolyte volume increase associated with the design?

3. What is the electronic conductivity of this porous MFC-Li/CNF electrode? It is known that Li₂S is one of solid electrolyte materials, and should have poor electronic conductivity. How much reduction of electronic conductivity of this electrode compared with Li metal? Does the larger overpotential at the later stage of long discharge for MFC-Li/CNF observed in Figure 5b associate with the poor electronic conductivity?

4. Do the authors have MFC-Li data on symmetric cells and Li-S full-cell? It will be helpful to show MFC-Li data so we can differentiate the positive effects from either the porous S/C structure or from the protective coatings of LiF, and Li₃Bi.

5. What is the C.E. for Li-S full cell? In figure 5a-b, at 1 mA/cm², the C.E. seems to be larger than 100%, please confirm.

6. RT long-term cycling data are missing. Please provide RT cycling data for comparison.

7. Carbon cloth as sulfur host usually only has smaller loading. Please describe more details on the methodology of the fabrication of high sulfur loading carbon cloth.

Reviewer #2 (Remarks to the Author):

This work claims to fabricate a porous lithium electrode which is protected with LiF and Li₃Bi Layer and is in direct contact with the solid electrolyte layer. The authors showed some nice results and

interesting cycling data. However, After carefully going through the manuscript, I regret to say that I won't recommend this work for publishing in Nature Communication keeping the lack of novelty and experimental support provided in this work.

Few comments

1. This work seems to be based on the previously published research work by Yi Cui (DOI: 10.1021/nl100504q | Nano Lett. 2010, 10, 1486–1491) utilizing exactly the same procedure for creating lithiated electrode which is not even cited by the authors.
2. Here, authors are claiming to form lithium electrode but the reaction of S with molten Li lead to the formation of lithium sulfide which settled to create the Li_2S rich surface and excess molten lithium attached to carbon created Li-rich surface. This should be supported by an experimental evidence like elemental mapping showing that Li-rich surface doesn't contain any residue of sulfur.
3. The dependency of lithium insertion on the amount of initial sulfur loading needs to be investigated as it will reflect on the solid electrolyte thickness as well as the thickness of the molten lithium.
4. Structural and quantitative investigation of LiF and Li_3Bi after cycling should also be investigated using NMR, TEM or any other technique in order to show that the layer remained stable in the stringent battery environment.

Response to the first reviewer

General comment: The work of “Rational design of spontaneous reactions to form a protected porous Li electrode for a stable Li-S battery” by Ren et al. has been reviewed carefully. In the work, the authors developed two steps of Li anode engineering that involved Li reaction with a porous S/C structure and BiF_3 , in which the protected Li electrodes show good cycling performance. However, there are several points listed below as reviewer’s comments that need to be clarified first. Although the work was characterized well, there are many similar works already published. The reviewer suggests the authors to provide more systematic and quantitative comparison between this work and previous published work to see where this work stand out.

Response: We thank the reviewer for the constructive suggestions. We would like to point out that compared to many other studies on Li protection, the key innovation of our work is in the spontaneous fabrication processes. In the response below, we follow the suggestions and compare our work systematically to others’ to show that we have achieved a scalable, solution-based processes towards a full protection of a porous Li scaffold in a Li-S battery, which has not been attained in the literature.

Comment 1: There are many works done on protecting Li for Li-S chemistry with 3D porous scaffolds, e.g. the work by Lin et al. (Nature Nanotechnology volume12, pages194–206 (2017)), and some works on forming protective layers with electro-polymerization e.g. the work by Wang et al., (DOI: 10.1021/acsami.8b07248). Therefore, this work is not with high novelty. The reviewer suggests the author to systematically compare their design and outcome to the already published work to see if this work standouts from previous works.

Response: We appreciate the suggestion and provide the following comparison,

which we wish could show our work's novelty. First, the two papers mentioned by the reviewers were indeed fine attempts of transformative impacts on the field of Li protection. However, we would argue that they would both fell short in providing an adequate protection for Li to survive hundreds of cycles in a Li-S battery. Specifically, Lin et al. (Nature Nanotechnology volume12, pages194–206 (2017)) thermally infused molten Li into layered graphene oxide (GO) paper for a composite of Li and reduced graphene oxide (Li/rGO) ¹. The as-fabricated electrode, however, has no protection against polysulfides in a Li-S battery. Wang et al. (DOI: 10.1021/acsami.8b07248) employed the polymerized 1,3-dioxolane to shield the planar Li ². With this protection, however, the battery still lost 37.5% of its capacity after 100 cycles at 0.1 C, as polymeric protective layers often suffer from polysulfide permeation and dendrite penetration.

As suggested by the reviewer, we compare our work to representative papers in the literature on the protection of porous Li electrodes (Table R1), with an emphasis on their effectiveness in achieving stable cycling in a Li-S battery, which is the ultimate goal of our work. Among these efforts, Liu et al. further employed inorganic solid electrolyte pellet based on Garnet-type $\text{Li}_{6.5}\text{La}_3\text{Zr}_{0.5}\text{Ta}_{1.5}\text{O}_{12}$ (LLZTO) for protecting the Li/rGO electrode ³. Xu et al. attempted to thinner the surface solid electrolyte layer by building a bilayer dense-porous structure ⁴. Though effective for the suppression of side reactions and dendrite, the use of inorganic solid electrolyte either as the surface protective layer or the scaffold persistently suffers from the fragility, the high Li^+ transport resistance, and the relatively low volumetric energy density.

In this work, instead of using the Garnet-type solid electrolytes, we turned to use the amorphous $\text{Li}_2\text{S-P}_2\text{S}_5$ solid electrolyte for surface protection. Unlike the high sintering temperature required for Garnets (>1100 °C), the sulfide-based electrolyte

can be sintered at a low temperature (160 °C), without side reaction (with carbon) or Li melting. Thus the solid electrolyte can be in-situ formed onto the Li-coated CNF matrix as a surface layer. Supported by the compressible and flexible interwoven CNF matrix, the fragility issue of the solid electrolyte can be much addressed, leading to a thin/dense protective layer with a low Li⁺ transport resistance.

Table R1 The reported porous Li scaffolds and main issues for use in lithium-sulfur full battery.

Ref.	Scaffold/substrate	Surface protection	Artificial SEI	Capacity retention
Lin et al. ¹	rGO	N/A	N/A	Unknown for Li-S
Liu et al. ³	rGO	Dense garnet	N/A	Unknown for Li-S
Xu et al. ⁴	Porous garnet	Dense garnet	N/A	91.7%, 50 cycles
Lin et al. ⁵	rGO	LiF	LiF	89%, 100 cycles
This work	CNF	Li ₂ S-P ₂ S ₅	Li ₃ Bi/LiF	93%, 200 cycles

Ref.	Li loading and Li utilization	Other issues
Lin et al. ¹	Unknown for Li-S	excessive addition of liquid electrolyte
Liu et al. ³	Unknown for Li-S	high resistance; fragility of the solid electrolyte
Xu et al. ⁴	4.3 mg cm ⁻² , 36.6% (39 mA g ⁻¹)	high resistance; high scaffold weight
Lin et al. ⁵	4.14 mg cm ⁻² , 12.7%(161 mA g ⁻¹)	non-continuous surface protective layer
This work	3.42 mg cm ⁻² , 41.2%(248.5 mA g ⁻¹)	

Moreover, inside the porous Li scaffold, liquid electrolyte is usually necessary for wetting the Li surface to address the interphase problem. A robust artificial SEI for smoothing the Li deposits and suppressing the consumption of liquid electrolyte thus

would be necessary. Lin et al. treated the Li/rGO electrode in the Freon gas atmosphere to form the LiF-based artificial solid electrolyte interphase (SEI) across the electrode for protection⁵. There are also some other reported methods to derive LiF-rich artificial SEI as shown in Table R2. However main concerns are raised for the coating condition (vapor deposition and atomic layer deposition), the quality of the coating layer (solution deposition, electrolyte additive, and blade casting), and the ionic conductivity of LiF (defect-free LiF is highly insulating)⁶. As a result, few of them can be readily applicable for coating the artificial SEI inside the porous Li scaffold at this stage.

Table R2 The reported methods for LiF coating on the Li or Cu foil.

Methods	Coating mechanisms	Main issues
Vapor deposition ^{7,8}	Reacting Li with the fluoride-donating gas or fluorine gas	Complex experimental setup, safety concern
ALD ⁹	Reacting Li salt with fluoride-donating gas	Operating temperature higher than the melting point of Li
Electrolyte additive ^{10,11}	Using LiF decomposed from additives to passivate Li anode	Non-uniform and uncontrollable
Blade casting ¹²	Lithiation of NiF ₂ to form LiF	Thick coating layer (>20 μm)

To address the aforementioned concerns, we propose an innovative method to solvate the metal fluoride precursor (BiF₃), using P₂S₅ as a complexation agent, enabling the solution deposition of artificial SEI. In spite of the reaction between Li₂S and P₂S₅ to form the solid electrolyte, the interfacial reaction between Li and BiF₃ spontaneously occurred at room temperature, generating the biphasic Li₃Bi/LiF coating. Both compositions were uniformly mixed in the nanoscale inside the artificial

SEI as shown in the TEM images (Figs. 3(e, f)) and Fig. S16), which promised to improve the ionic conductivity. Owing to the abundant Li ion interstitial sites provided by Li_3Bi , Li^+ can facily transport across the interphase of $\text{Li}_3\text{Bi}/\text{LiF}$. Besides, Li_3Bi possesses a high Young's modulus (22.2 GPa) and an adequate ductility, improving the overall mechanical strength of the artificial SEI.

Overall, the porous Li scaffold and multiple protections were achieved through near room temperature melt infusions and solution-based reactions. The roles of each composition (LiF , Li_3Bi and $\text{Li}_2\text{S-P}_2\text{S}_5$ based solid electrolyte) were well defined, which proofed to alleviate the issues of dendrite and parasitic reactions. Owing to the rationally-designed electrode structures and compositions, for the as-developed Li-S full batteries, the Li anode could be cyclable with a relatively high Li utilization (Fig. R1a) and a relatively low capacity fading rate (Fig. R1b), compared with the open literatures.

Fig. R1 The specific capacity and specific current (calculated based on the mass of Li) achieved in literature and this work. The full batteries are all based on Li-S battery chemistry. The areal loadings of Li (based on the anode size) and the cycle numbers achieved are included in the figure (details can be found in Table R1). The open symbols represent the specific capacities at initial cycles, while the filled symbols represent the specific capacities after cycling. Some of the reported planar anodes such as the bare Cu foil, and the Cu foil protected by double-layered nanodiamond are

also included for comparison^{13,14}.

Comment 2: Keep in mind in Li-S battery, the ratio between electrolyte volume and sulfur loading (E/S) is an important factor (E/S should be low) for it to be useful. The design of porous Li electrode will need more electrolyte to wet the interfaces. Do the authors have estimation on how much porosity and electrolyte volume increase associated with the design?

Response: We agree with the reviewer that it is important to quantify the amount of liquid electrolyte needed in the porous Li electrode. We estimated the amount of Li infused inside the scaffold using a half-cell setup with a large-sized protected Li foil (16 mm in a diameter) as the counter electrode and a small-sized Li/S-CNF (5 mm in diameter) as the working electrode. Assuming complete Li stripping, we can estimate the amount of Li inside the composite ($3.68 \text{ mg}_{\text{Li}} \text{ cm}^{-2}$, the stripping curve could be found in Fig. S7f or Fig. R9f). The amount of Li_2S thus can be calculated. The detailed gravimetric and volumetric fractions of these electrode components are shown in Fig. R2. Porosity could be estimated as around 7.1 % for the Li/S-CNF electrode, which would be filled with the liquid electrolyte (around $0.6 \mu\text{L cm}^{-2}$). The weight ratio between liquid electrolyte and Li is as low as 0.16. For the MFC-Li/CNF electrode, the amount of liquid electrolyte needed could be estimated at the same level. The porosity of the electrode can be tuned by adjusting the density of carbon nanofibers as well as the initial sulfur loading.

Fig. R2 The calculation of gravimetric/volumetric fractions of electrode components

($\rho_{\text{Li}} = 0.534 \text{ g cm}^{-3}$, $\rho_{\text{Li}_2\text{S}} = 1.66 \text{ g cm}^{-3}$, $\rho_{\text{CNF}} = 2.25 \text{ g cm}^{-3}$, $\rho_{\text{electrolyte}} \approx 0.90 \text{ g cm}^{-3}$).

The Li-S/CNF electrode was compressed before characterization and testing. Before compression, the electrode is more porous and allows for the solution deposition to occur uniformly through it.

Comment 3: What is the electronic conductivity of this porous MFC-Li/CNF electrode? It is known that Li₂S is one of solid electrolyte materials, and should have poor electronic conductivity. How much reduction of electronic conductivity of this electrode compared with Li metal? Does the larger overpotential at the later stage of long discharge for MFC-Li/CNF observed in Figure 5b associate with the poor electronic conductivity?

Response : Thanks for raising this essential question. The electronic/ionic conductivities of the electrodes were compared by measuring the symmetric cells' voltages at a constant current. Li foil exhibits a negligible electronic resistance ($\sigma_{\text{electron}} = 1.08 \times 10^5 \text{ S cm}^{-1}$). When Li was infused inside the Li/S-CNF electrode, due to the insulating nature of Li₂S, the resistance rose to $4.52 \text{ } \Omega \text{ cm}^{-2}$. For MFC-Li/CNF, due to the formation of a more continuous surface solid electrolyte layer, the transport of electron across the electrode surface can be well insulated, leading to the highest

areal resistance of $14.9 \Omega \text{ cm}^{-2}$. The $\text{Li}_3\text{Bi}/\text{LiF}$ -based artificial SEI is also insulating as shown in the measurement result for MFC-Li ($8.93 \Omega \text{ cm}^{-2}$), which is beneficial for stabilizing the Li/electrolyte interphase.

Further, as indicated by the reviewer, during the long term discharge of the symmetric cell, in the MFC-Li/CNF electrode where Li was stripped, the electronic resistance would decrease as Li was much more conductive than carbon. While in the electrode where Li was deposited, there would be a rise of ionic resistance, as the expansion of Li deposit layer on the carbon fiber would also lead to the expansion of the electrode and the decrease of the electrode porosity. Both of the decrease of electronic and ionic conductivity would lead to the sloping discharging curve for the symmetric cell as shown in Fig. 5b. Finally, when Li was largely stripped, the discharging curve would undergo a sudden decrease.

Fig. R3 Symmetric cells' polarization curves. For testing MFC-Li/CNF, MFC-Li and Li/S-CNF and Li, two electrodes are stacked, with their Li-rich sides attached on the stainless steels. No liquid electrolyte was added for wetting the electrode.

Comment 4: Do the authors have MFC-Li data on symmetric cells and Li-S full-cell? It will be helpful to show MFC-Li data so we can differentiate the positive effects

from either the porous S/C structure or from the protective coatings of LiF, and Li₃Bi.

Response: We agree with the reviewer that it is important to compare MFC-Li/CNF with MFC-Li. The symmetric cell with the MFC-Li electrode was charged and discharged with an areal capacity of 1.0 mAh cm⁻² at a current density of 1.0 mA cm⁻² or 2.5 mA cm⁻², respectively. Stable operation without voltage hysteresis lasted for 300 cycles (600 hours) during 1.0 mA cm⁻² cycling, considerably better than the bare Li metal as shown in Fig. R4. However, for 2.5 mA cm⁻², within 300 cycles, the overpotential of the symmetric cell with the MFC-Li electrode promoted to over twice of its initial value as shown in Fig. R5. This can be attributed to the fact that the SEI layer thickens during cycling, which would block the ion transport across the interphase.

Fig. R4 (a) Representative cycling voltage profiles at 1.0 mAh cm⁻² at 1.0 mA cm⁻² for symmetric cell assembled with the MFC-Li electrode; (b) the average cell voltage versus cycle index; (c) voltage profiles at different current densities for the symmetric

cell with the MFC-Li electrode.

Fig. R5 (a) Representative cycling voltage profiles at 1.0 mAh cm^{-2} at 2.5 mA cm^{-2} for symmetric cell assembled with the MFC-Li electrode; (b) the average cell voltage versus cycle index.

Further, for Li-S battery testing, we have operated the MFC-Li anode under a low sulfur loadings (2.8 mg cm^{-2}). Its performance at a 0.5 mA cm^{-2} was relatively stable (0.134% decay per cycle), but it suffered from a relatively fast decay at 2.0 mA cm^{-2} (0.355% decay per cycle), due to the significant deformation of the Li/electrolyte interphase as shown in Fig. R6a. Under the same sulfur loading, the specific capacity achieved by MFC-Li/CNF was remarkably higher than the MFC-Li, due to the decrease of overpotential in the anode side, as shown in Fig. R6b. Through the comparison, we can confirm that combining the protective coating and porous Li scaffold is essential for stabilizing the Li/electrolyte interphase.

Fig. R6 (a) Cycling performances of Li-S batteries assembled with MFC-Li or bare Li anodes at 0.5 and 2.0 mA cm⁻² (based on the area of cathode) with an areal sulfur loading of 2.8 mg cm⁻²; (b) voltage profiles for the batteries with MFC-Li or MFC-Li/CNF anode.

Comment 5: What is the C.E. for Li-S full cell? In figure 5a-b, at 1mA/cm2, the C.E. seems to be larger than 100%, please confirm.

Response: Thanks for this constructive comment. In Fig. 6c, we showed both charge and discharge capacities. In the revised manuscript, we further included the coulombic efficiency in Fig. 6c (as can be also seen in Fig. R7). At 45 °C, cycling of Li-S full batteries achieved an average CE value of 99.7% at 1.0 mA cm⁻², and 96.8% at 4.0 mA cm⁻², for 200 cycles. There were not significant evolutions of CE values during cycling.

Comment 6: RT long-term cycling data are missing. Please provide RT cycling data for comparison.

Response: Thanks for this very helpful comment. We have operated the batteries at room temperature for comparison before. For clear viewing, we didn't include them in the manuscript. The average capacity fading rates were 0.033% and 0.092% for 23 °C cycling at 1.0 mA cm⁻² and 4.0 mA cm⁻², respectively. While the average capacity fading rates were 0.035% and 0.123% for 45 °C cycling at 1.0 mA cm⁻² and 4.0 mA cm⁻², respectively (Fig. R7). At a higher temperature, the thickening of native SEI will be accelerated, leading to the slight increase of capacity fading, as shown in our test results.

Fig. R7 Performance of a Li-S battery employing the MFC-Li/CNF electrode. (a, b) Voltage profiles of Li-S batteries assembled with MFC-Li/CNF electrode with a sulfur loading of 6.8 mg cm⁻² (theoretical areal capacity 11.4 mAh cm⁻²) at 23 °C (a); and at 45 °C (b); (c) cyclability of full batteries at 1.0 mA cm⁻² or 4.0 mA cm⁻², at 23 °C or 45 °C.

Comment 7: Carbon cloth as sulfur host usually only has smaller loading. Please describe more details on the methodology of the fabrication of high sulfur loading carbon cloth.

Response: We appreciate this very constructive suggestion from the reviewer. Specifically for the purpose of performance evaluation, we screened out the pristine carbon cloth as the cathode current collector, which is of a low tortuosity, and a robust interwoven structure. The untreated carbon fiber in the carbon cloth is highly

graphitic. As a result, there would be minor structural or compositional change for the carbon cloth over cycling. Surface engineering (e.g. decoration of nanostructured polysulfide binding materials) of the electrode can enhance the sulfur utilization, however, there presumably would be the clogging of micropores or passivation of the active sites during cycling. So herein carbon cloth electrode was not subjected to any pretreatment. The sulfur/carbon cloth electrode with a high loading was reported before by our group and other researchers. For example, with an activated carbon cloth, Elazari et al. attained an areal loading of sulfur around 6.5 mg cm^{-2} at 0.98 mA cm^{-2} , reaching a maximum specific capacity of 1050 mAh g^{-1} . Though their carbon cloth possessed abundant micropores and there was intensive addition of LiNO_3 (2.0 wt%) as the side reaction inhibitor, their battery's fading rate was still relatively fast (0.25% decay per cycle for 80 cycles), indicating the necessity of rationally modifying the Li anode. We have included the corresponding discussion in the main context.

We sincerely thank the reviewer for the constructive comments, which are really helpful for improving the quality of this paper!

Response to the second reviewer

General comment: This work claims to fabricate a porous lithium electrode which is protected with LiF and Li_3Bi Layer and is in direct contact with the solid electrolyte layer. The authors showed some nice results and interesting cycling data. However, after carefully going through the manuscript, I regret to say that I won't recommend this work for publishing in Nature Communication keeping the lack of novelty and experimental support provided in this work.

Response: Firstly, we would like to thank the reviewer for carefully reviewing our manuscript and acknowledging the performance achieved in this study. We also

understand the reviewer's concern regarding to the novelty and experiment support. We sincerely hope that we could address the reviewers' concerns from the following perspectives.

Comment 1: This work seems to be based on the previously published research work by Yi Cui (DOI: 10.1021/nl100504q | Nano Lett. 2010, 10, 1486–1491) utilizing exactly the same procedure for creating lithiated electrode which is not even cited by the authors.

Response: We thank the reviewer for raising this concern. The previous work (New nanostructured Li_2S /silicon rechargeable battery with high specific energy, DOI: 10.1021/nl100504q | Nano Lett. 2010, 10, 1486–1491) proposed a silicon/ Li_2S full battery configuration, in which an organic lithium superbase (n-butyllithium) was employed for the prelithiation of sulfur/carbon composite. The previous work was impressive, however when preparing the manuscript, we didn't realize it was strongly correlated with our work. Much different from the previous work, the motivation of our work is to create a protected porous Li anode, which can be deeply cyclable in the Li-S full battery. We further try to illustrate the strength of this work, by systematically comparing our work with the literature, as could be found in our response to the first question of the first reviewer.

Comment 2: Here, authors are claiming to form lithium electrode but the reaction of S with molten Li lead to the formation of lithium sulfide which settled to create the Li_2S rich surface and excess molten lithium attached to carbon created Li-rich surface. This should be supported by an experimental evidence like elemental mapping showing that Li-rich surface doesn't contain any residue of sulfur.

Response: We appreciate the valuable suggestion from the reviewer. The Li_2S -rich

surface was formed from the gravitational setting of Li_2S during the formation process of Li-S/CNF composite. After the reaction and Li infusion, the CNF can be largely lithiated, which possesses a much improved lithiophilicity^{15,16}. The lithiated CNF thus can trap the molten Li to form a fibrous network. In contrast, Li_2S has poor affinity with both carbon and Li, so it gradually condenses on the electrode's bottom to form a Li_2S -rich surface.

The formation of Li_2S -rich surface allows us to create a conformal solid electrolyte layer to shut down the polysulfide crossover. We support this claim by providing the elemental mapping of the electrode's cross section, and a gradient distribution of sulfur can be visualized as shown in Fig. R8(a, b). Despite the cross section, the surface SEM also showed the designed Li_2S -rich surface exhibited a dense morphology due to the condensation of Li_2S and the surface sulfur fraction (atomic) reached 79% based on the EDX mapping result (Fig. R8(c, d)). On the designed Li-rich surface, we further observed the Li-coated fibers with micrometer diameter, with an atomic fraction of sulfur as low as 6% (Fig. R8(e, f)).

Fig. R8 (a, b) Cross-sectional SEM of the Li/S-CNF (a), and its EDX (b); (c, d) SEM of the so-called Li_2S -rich surface of Li/S-CNF (c), and its EDX (d); (e, f) SEM of the so-called Li-rich surface of Li/S-CNF (e), and its EDX (f). The atomic fraction of sulfur was calculated considering sulfur, carbon and oxygen.

Comment 3: The dependency of lithium insertion on the amount of initial sulfur loading needs to be investigated as it will reflect on the solid electrolyte thickness as well as the thickness of the molten lithium.

Response: We thank the reviewer for raising this very constructive question. As suggested by the reviewer, the initial loading of sulfur can have a significant impact on the structural property of the Li electrode. At a higher sulfur loading, the exothermic lithiation reaction can provide more heat to reduce the viscosity of molten Li and drive the wetting of Li on the carbon surface. From the cross-sectional SEM in Fig. R9, we can observe a higher initial sulfur loading results in a denser filling of the

CNF matrix and larger volumetric expansion. We further estimated the amount of Li infused inside the matrix using a half-cell setup with a large MFC-Li (16 mm in a diameter) as the counter electrode and a small Li/S-CNF (5 mm in diameter) as the working electrode, The corresponding voltage profiles are shown in Fig. R9(c, f, i, l). Assuming complete Li stripping, the volume fraction taken up by Li can be derived. Herein, we specifically chose an initial sulfur loading of 2.5 mg cm^{-2} , which was almost the minimum loading value to trigger the uniform Li filling (Fig. R10). Besides, under this condition, the volume expansion ratio of the anode can be also maintained at a reasonable value ($\sim 70\%$).

Fig. R9 Cross-sectional SEMs of Li-S/CNF electrodes fabricated with different initial sulfur loadings and their delithiation (0.5 mA cm^{-2}) voltage profiles. Sulfur loading: (a-c) 1.0 mg cm^{-2} ; (d-f) 2.5 mg cm^{-2} ; (g-i) 4.0 mg cm^{-2} ; (j-l) 8.0 mg cm^{-2} . When charging up to 2.1 V, we can further observe the voltage profiles representing the

delithiation of Li_2S .

Fig. R10 The impacts of the initial sulfur loading on the volume fraction of Li and the volume expansion ratio of electrode.

Comment 4: Structural and quantitative investigation of LiF and Li_3Bi after cycling should also be investigated using NMR, TEM or any other technique in order to show that the layer remained stable in the stringent battery environment.

Response: We sincerely thank the reviewer for this very constructive comment. Both structural and compositional analysis were conducted. There was negligible electrode level deformation after high rate cycling (4.0 mA cm^{-2}) as shown in Fig. R11(a, b). From XPS depth profiling (Fig. R11c), we observed the coverage of a native passivation layer (SEI) on the solid electrolyte surface, likely formed from the decomposition of LiTFSI. If we further probed inside the solid electrolyte, we observed that the peaks of native passivation layer diminished, indicating that the surface solid electrolyte layer prevented the permeation of liquid electrolyte. On the Li-rich surface, we could observe the Li-coated fiber maintained its morphology over cycling (Fig. R11(d, e)). XPS depth profiling further indicated the Li-coated fibers were passivated by both the $\text{Li}_3\text{Bi}/\text{LiF}$ -based protective layer and the native SEI (Fig. R11f). These results testify that the structure of the porous electrodes and the

compositions of protective layers ($\text{Li}_2\text{S-P}_2\text{S}_5$ based solid electrolyte and $\text{Li}_3\text{Bi/LiF}$ based artificial SEI) can well retain over cycling.

Fig. R11 (a, b) Cross-sectional SEM (a) and top surface SEM (b) of the MFC-Li/CNF after 200 cycles (4.0 mA cm^{-2}) in the Li-S battery (6.8 mg cm^{-2} sulfur loading). (c) XPS depth profiling (S 2p) for the top surfaces of MFC-Li/CNF. (d, e) Bottom surface SEM of the MFC-Li/CNF after cycling. (f) XPS depth profiling (Bi 4f) for the bottom surfaces of MFC-Li/CNF. The ranges of S 2p and Bi 4f spectra are overlapped. The Ar sputter speed was 13 nm min^{-1} .

We sincerely thank the reviewer for the constructive comments, which are really helpful for improving the quality of this paper!

References

1. Lin, D. *et al.* Layered reduced graphene oxide with nanoscale interlayer gaps as a stable host for lithium metal anodes. *Nat. Nanotechnol.* **11**, 626 (2016).
2. Wang, Y. *et al.* Electrochemically Controlled Solid Electrolyte Interphase Layers Enable Superior Li-S Batteries. *ACS Appl. Mater. Interfaces* **10**,

- 24554–24563 (2018).
3. Liu, Y. *et al.* Transforming from planar to three-dimensional lithium with flowable interphase for solid lithium metal batteries. *Sci. Adv.* **3**, eaao0713 (2017).
 4. Xu, S. *et al.* All-in-one Lithium-Sulfur Battery Enabled by a Porous-Dense-Porous Garnet Architecture. *Energy Storage Mater.* (2018).
 5. Lin, D. *et al.* Conformal lithium fluoride protection layer on three-dimensional lithium by nonhazardous gaseous reagent freon. *Nano Lett.* **17**, 3731–3737 (2017).
 6. Zhang, Q. *et al.* Synergetic effects of inorganic components in solid electrolyte interphase on high cycle efficiency of lithium ion batteries. *Nano Lett.* **16**, 2011–2016 (2016).
 7. Lin, D. *et al.* A conformal lithium fluoride protection layer on three-dimensional lithium by non-hazardous gaseous reagent Freon. *Nano Lett.* (2017).
 8. Zhao, J. *et al.* Surface fluorination of reactive battery anode materials for enhanced stability. *J. Am. Chem. Soc.* **139**, 11550–11558 (2017).
 9. Xie, J. *et al.* Stitching h-BN by atomic layer deposition of LiF as a stable interface for lithium metal anode. *Sci. Adv.* **3**, eaao3170 (2017).
 10. Lu, Y., Tu, Z. & Archer, L. A. Stable lithium electrodeposition in liquid and nanoporous solid electrolytes. *Nat. Mater.* **13**, 961–969 (2014).
 11. Suo, L. *et al.* Fluorine-donating electrolytes enable highly reversible 5-V-class Li metal batteries. *Proc. Natl. Acad. Sci.* 201712895 (2018).
 12. Peng, Z. *et al.* Stabilizing Li/electrolyte interface with a transplantable protective layer based on nanoscale LiF domains. *Nano Energy* **39**, 662–672 (2017).

13. Nanda, S., Gupta, A. & Manthiram, A. A Lithium–Sulfur Cell Based on Reversible Lithium Deposition from a Li₂S Cathode Host onto a Hostless-Anode Substrate. *Adv. Energy Mater.* 1801556 (2018).
14. Liu, Y. *et al.* An Ultrastrong Double-Layer Nanodiamond Interface for Stable Lithium Metal Anodes. *Joule* (2018).
15. Shao, Y. *et al.* Drawing a Soft Interface: An Effective Interfacial Modification Strategy for Garnet-Type Solid-State Li Batteries. *ACS Energy Lett.* **3**, 1212–1218 (2018).
16. Liu, S. *et al.* Large-scale synthesis of high-quality lithium-graphite hybrid anodes for mass-controllable and cycling-stable lithium metal batteries. *Energy Storage Mater.* **15**, 31–36 (2018).

Reviewers' comments:

Reviewer #1 (Remarks to the Author):

This manuscript is much improved by replying reviewers' comments adequately. The major issue for this work is the lack of novelty. The protection layer forms by a spontaneous reaction is not a new idea, and the metal fluoride protection idea has been published several times in the literature. The work provides good insights on exploring a specific reaction that serves the purpose, which should be published elsewhere. But I won't recommend its publication in Nature Communication.

Reviewer #2 (Remarks to the Author):

After going through the significant efforts by the authors, I am satisfied with the revisions and recommend the article for publishing.

Reviewer #3 (Remarks to the Author):

The authors present a multistep procedure for the protection of Li anode in Li-S battery, indeed a key for enabling high Coulombic efficiency and long-term cyclability in practical Li-S batteries. The approach is rather novel resulting in a pretty reasonable performance in the absence of LiNO₃ in the electrolyte. I can recommend it for publication in Nat Comm if the authors demonstrate that amongst those approaches which deal with the ex-situ formation of protective layers on Li metal their approach is the best. They should be able to that by comparing the 1) Areal capacity of different test cells and 2) the presence/absence of LiNO₃. Reason being that the suitability of any sort of in-situ/ex-situ protection layer can be best examined in the absence of LiNO₃ and at practically high gravimetric/areal capacities.

1- If I got it right (please confirm), the presented cycling data in this work is in the absence of LiNO₃ which is impressive. In that case the authors should highlight that and compare their work with recent literature such as NATURE NANOTECHNOLOGY | VOL 13 | APRIL 2018 | 337–344 "2D MoS₂ as an efficient protective layer for lithium metal anodes in high-performance Li–S batteries" where LiNO₃ is used in the electrolyte and elucidating the effect of the protection layer is rather difficult.

2- The authors mention that their "protected porous Li electrode enables stable operation of a Li-S battery of a high sulfur loading (6.8 mg cm⁻²) at a high current density (4.0 mA cm⁻²) for more than 200 cycles" and "At 23 °C, the discharge capacity achieved at a cathode current density of 4.0 mA cm⁻² (anode current density 3.4 mA cm⁻²) was 4.14 mAh cm⁻²". Again, while this result is impressive in the absent of LiNO₃, 4.14 mAh cm⁻², is not going to be a practical areal capacity for the future Li-S battery. The math is simple- the areal capacity of the graphite anode in commercial LIB is 2.5-3.5 mAh cm⁻² and for a Li-S battery with a lower operating voltage (≈ 2.1 V compared to ≈ 3.7 of that of LIB) to rival LIB, the areal capacity should be at least 6 mAh cm⁻². Given the excess of pretty much every single component in the Li-S battery (for example you are using 100 μ m liquid electrolyte, additional protection layers, a carbon cloth current collector), even at 6 mAh cm⁻² and gravimetric capacities above 1000 mAh g⁻¹s, translation to LIB is questionable. I suggest avoid using terms such as high current density when both the areal and gravimetric capacity are not satisfactory compared to the practical levels.

3- Even though 6.8 mg cm⁻² is certainly a high sulfur loading, a gravimetric capacity of 800 mAh g⁻¹s at such a low current density (1 mAh cm⁻²) means that two critical issues of polysulfides shuttle and

Li anode stress are not truly examined in these batteries, the latter in particular. I suggest assembling test cells with high areal capacity cathodes ($> 6 \text{ mAh cm}^{-2}$) over a few tens of cycles to ensure true examination of the full range of technical challenges that are present in a practical Li-S battery. Perhaps use a higher sulfur-loading cathode and monitor the performance at room temperature while areal capacity is higher than 6 mAh cm^{-2} . And please plot the Coulombic efficiency data in a more visible fashion, perhaps magnify it in the range of 90 to 100 %.

4- Given that your approach is a two-step process, have you conducted cycling tests with 1) just porous Li electrode, comprising fibrous networks of a Li shell and a carbon core with no coating on the surface, and 2) a lithium foil coated with your proposed composite layer of Li_3Bi and LiF (if its formation is at all possible on a bare lithium foil). Also, it would be interesting to examine the performance in the presence of LiNO_3 additive in the electrolyte.

5- The authors should present discussions and argument on why they think their approach is a better alternative for the simple and commonly used approach of using LiNO_3 as the electrolyte additive. As far as I'm concerned, the literature of Li-S is overwhelmed with similar performance metrics as the authors present in this work where an in-situ SEI layer is formed on the Li metal by using LiNO_3 or other additives in the electrolyte composition.

In summary, I find the approach interesting and novel. Battery manufacturers are not going to be fond of formation of ex-situ protection layers or composite Li electrode given the complexity of the procedures and the demand for the whole process to be conducted under Ar. Nonetheless, I believe this work would be of the interest to the scientific community. The results even though not impressive in the general literature of Li-S can be considered as very good given the absence of LiNO_3 . To ensure the true suitability of this Li coating, test cells must be configured with high areal capacity cathodes to allow for passage of realistic current densities.

Response to reviewers' comments

Reviewer #1 (Remarks to the Author):

This manuscript is much improved by replying reviewers' comments adequately. The major issue for this work is the lack of novelty. The protection layer forms by a spontaneous reaction is not a new idea, and the metal fluoride protection idea has been published several times in the literature. The work provides good insights on exploring a specific reaction that serves the purpose, which should be published elsewhere. But I won't recommend its publication in Nature Communication.

Response: We appreciate the comments by the reviewer, which has helped improve this manuscript tremendously. With the points elaborated below, we hope to clear the doubt over the novelty of the work.

First, with respect to the outstanding work on both spontaneous reactions and protective metal fluoride formation, the significance of our work is that we have *addressed the challenges in combining these two promising approaches to solve the issues in Li-S batteries*. No previous spontaneous reaction was able to meet all the needs for Li protection in a Li-S battery. For example, the reactions between Li and a solid precursor resulted in too thick a protective layer¹. Reactions with a gas, e.g. Freon or fluorine, though effective in protecting porous Li^{2,3}, brought hazard and high cost to the fabrication. Spontaneous reactions at the liquid/solid interface are arguably the most promising approach with many recent progresses as this manuscript is going through the review process. For example, Pang et al. used InF₃ as a stabilizing additive in the electrolyte⁴; Yan et al. solvated CuF₂ together with LiNO₃, as the precursor for Li surface treatment⁵, and Lang et al. used PVDF-DMF solution for LiF coating on planar Li metal⁶. However, none of these reactions could generate a sufficiently robust and conductive layer that is also immune to the attack by polysulfides; InF₃ has a rather low

solubility; CuF_2 produces Cu which is unstable towards polysulfides, and DMF, though initiating the LiF formation, is known to attack Li⁷⁻⁹.

In this work, we present *a new design strategy that builds upon the need for all the ingredients to work together before and after the spontaneous reaction*. After the reaction, we need LiF, the most effective component in SEI for Li protection, but given the brittleness of LiF, we also need a stronger component to form a composite, likely a Li alloy. We need another layer on top to prevent the attack of polysulfide, likely a solid electrolyte, and this protection has to go onto a porous Li electrode for a practical areal loading. Meeting all these requirements are challenging, but intriguingly, they also bring us more ingredients, which opened up the space of the reaction design. Instead of using LiF directly, which is notoriously insoluble for a solution-phase reaction, we exploited Bi as needed in the alloy component, in the form of BiF_3 as the fluoride source. The solvation of BiF_3 required P_2S_5 , a complexant known for forming solid electrolytes when reacting with Li_2S ¹⁰. Sulfur for the Li_2S then became the perfect ‘lubricant’, whose affinity to carbon matrix and exothermic reaction with Li marry the two. Such a design rationale eventually led to the good battery performance, the strongest evidence is that the work is a viable solution to the Li protection in Li-S batteries.

Second, we would like to argue that the present work provides more than a specific reaction, but a general approach towards an effective metal protection in alkaline metal batteries by offering the following mechanistic insights:

(a) *Metal fluoride solvation*. The solvation of BiF_3 with P_2S_5 through Lewis acid-base interaction, as confirmed by the ³¹P spectra of NMR, can be a general strategy for solvating other metal fluorides as shown in Fig. R, and it open up a new opportunity for forming protective coating via a solution-phase spontaneous reaction on a wide range of anodes, including metallic Na, K, and alloys. We have now included the following figure into the Supporting Information to prove the generality of the solvation

mechanism for two more metal fluoride (Fig. R1), and to show the general feasibility of the corresponding coating strategy (Fig. R2).

Fig. R1 (a, b) Photography of the mixtures of (a) $\text{SbF}_3\text{-P}_2\text{S}_5$ and (b) $\text{InF}_3\text{-P}_2\text{S}_5$ with different molar ratios (1:0, 1:0.25, 1:0.5, 1:1, 1:2, 0:1, from left to right, all with 50 mM metal fluoride) in DME. The inset figures show the supernatant solution with a 1:1 molar ratio of metal fluoride: P_2S_5 . (c) Photography showing the spontaneous reaction between Li and InF_3 (50 mM) solvated with P_2S_5 in DME (upper row), and a control experiment without P_2S_5 in the solution (lower row). The surface passivation of Li can complete after 2-hour soaking at 45 °C. The coated side of Li foil was turned for taking

photo as shown in the inset figure. *These figures could be found in Fig. S27 and Fig. S28.*

Fig. R2 (a) XRD patterns for the as-prepared $\text{InF}_3\text{-P}_2\text{S}_5$ coated Li electrode; (b) SEM image of the cross section of the $\text{InF}_3\text{-P}_2\text{S}_5$ coated Li electrode. *This figure could be found in Fig. S29.*

(b) *The ionic conductivity and the microstructure of the artificial SEI.* Our experimental and computational studies have revealed the benefits of the surface layer as artificial SEI. The TEM shows that the high conductivity of our $\text{Li}_3\text{Bi/LiF}$ -based composite-SEI likely originated from the uniform distribution of crystalline grains of LiF and Li_3Bi at a nanoscale along the in-plane direction (Fig. 3(d-f)). We also used density functional theory to reveal that the low energy barrier of surface diffusion would promote uniform Li deposition through the artificial SEI. To reveal more structural details, we have now characterized the SEI layer with HR-TEM, confirming nanometer-sized crystallites of LiF and Li_3Bi (Fig. R3). We have also quantified the value of the ionic conductivity of the surface layer with a careful set of experiments given its critical role in stabilizing the anode (Fig. R4). We first combined cross-sectional SEM and TOF-SIMS to measure the thickness of a $\text{Li}_3\text{Bi/LiF}$ layer on the surface of Li metal (Fig. R4(a, b)), and then measured its ionic conductivity in a dry, symmetric cell to be $6.9 \times 10^{-4} \text{ S cm}^{-1}$ (Fig. R4c), much higher than defect-free LiF ($\sim 10^{-12} \text{ S cm}^{-1}$)¹¹. This result

corroborates the suggested benefits of our design strategy.

Fig. R3 (a) HR-TEM image of the $\text{Li}_3\text{Bi}/\text{LiF}$ -based artificial SEI; (b) its enlarged image with lattices indexed. *This figure could be found in Fig. S16.*

Fig. R4 (a) SEM image of the cross section of the MFC-Li electrode with a $5\text{-}\mu\text{m}$ $\text{Li}_3\text{Bi}/\text{LiF}$ surface layer; (b) TOF-SIMS profile of the MFC-Li surface. (c) EIS profiles of the symmetric cell with the MFC-Li electrodes (without liquid electrolyte and separator). For the measurement, two MFC-Li electrodes were stacked in CR-2032 coin cell, with their Li-rich sides attached on the stainless steels and their $\text{Li}_3\text{Bi}/\text{LiF}$ layers in direct contact. No liquid electrolyte was added for wetting the electrode. *This figure could be found in Fig. S13.*

At last, we would like to thank the reviewer again for the reminder of previous work on spontaneous reactions and metal fluoride coating. Although we had a summary

in Introduction on metal fluoride coating, we have now revised Introduction to give a review on liquid-phase spontaneous reactions, and discuss *how our approach differ from the preceding work*.

Reviewer #2 (Remarks to the Author):

After going through the significant efforts by the authors, I am satisfied with the revisions and recommend the article for publishing.

Response: We sincerely thank the reviewer for the kind acknowledgement and we will move on to carefully revise the manuscript.

Reviewer #3 (Remarks to the Author):

The authors present a multistep procedure for the protection of Li anode in Li-S battery, indeed a key for enabling high Coulombic efficiency and long-term cyclability in practical Li-S batteries. The approach is rather novel resulting in a pretty reasonable performance in the absence of LiNO₃ in the electrolyte. I can recommend it for publication in Nat Comm if the authors demonstrate that amongst those approaches which deal with the ex-situ formation of protective layers on Li metal their approach is the best. They should be able to that by comparing the 1) Areal capacity of different test cells and 2) the presence/absence of LiNO₃. Reason being that the suitability of any sort of in-situ/ex-situ protection layer can be best examined in the absence of LiNO₃ and at practically high gravimetric/areal capacities.

Response: We are grateful to the reviewer for the constructive suggestions. In the response below, we follow the reviewer's suggestions and compare our work systematically to others to show that we have achieved a deeply cyclable Li anode with a high areal capacity in the Li-S full battery, in the absence of LiNO₃ electrolyte additive.

1. If I got it right (please confirm), the presented cycling data in this work is in the absence of LiNO_3 which is impressive. In that case the authors should highlight that and compare their work with recent literature such as NATURE NANOTECHNOLOGY | VOL 13 | APRIL 2018 | 337–344 “2D MoS_2 as an efficient protective layer for lithium metal anodes in high-performance Li–S batteries” where LiNO_3 is used in the electrolyte and elucidating the effect of the protection layer is rather difficult.

Response: As the reviewer noticed, we operated our battery in a classical ether-based electrolyte without LiNO_3 . Following the suggestion, we have now made a systematic comparison between our work and recent literature using LiNO_3 for Li-S full batteries, as shown in Table. R1. The comparison shows that the as-developed Li-S full batteries (sulfur loading 6.8 mg cm^{-2} or 10.2 mg cm^{-2}) indeed delivered outstanding performance in terms of the areal capacity, current density, the utilization of Li and the capacity decay. We noticed that Liu et al.’s and Cha et al.’s work achieved higher cycle numbers^{12,13}. However, the areal capacity, current density and the Li utilization ratio of their work were lower. At a higher current density, for example, the use of LiNO_3 may not bring the same benefit as we will show in the response to the 4th comment of the reviewer’s.

Table R1 A performance comparison between our Li-S battery and those using LiNO₃ as electrolyte additives.

Ref.	Substrate	Surface protection	LiNO ₃ conc.	Capacity/current	Utilization	Cycle life	Retention
Nanda et al. ¹⁴	Cu	N/A	0.2 M	0.84 mAh cm ⁻² (0.84 mA cm ⁻²)	17.9%	100	51.5%
Liu et al. ¹²	Cu	DND-polymer	0.3 M	0.90 mAh cm ⁻² , (1.25 mA cm ⁻²)	18%	400	67%
Chen et al. ¹⁵	Li foil	Alucone	0.3 M	3.5 mAh cm ⁻² , (0.95 mA cm ⁻²)	2.9%	140	94%
Cha et al. ¹³	Li foil	MoS ₂	0.25 M	3.8 mAh cm ⁻² , (2.9 mA cm ⁻²)	15.4%	1200	85%
Tang et al. ¹⁶	Li foil	Li _x Si	0.1 M	1.82 mAh cm ⁻² , (1.675 mA cm ⁻²)	1.5%	160	91%
Liu et al. ¹⁷	Co/N-PCNS	N/A	0.15 M	1.2 mAh cm ⁻² , (0.33 mA cm ⁻²)	40%	60	68%
Cai et al. ¹⁸	TCF	Doped carbon shell	0.3 M	4.0 mAh cm ⁻² , (2.73 mA cm ⁻²)	30%	200	77.5%
Chang et al. ¹⁹	Carbon cloth	Nano Cu	0.3 M	2.4 mAh cm ⁻² , (2.0 mA cm ⁻²)	40%	250	72.5%
This work	CNF	Li ₂ S-P ₂ S ₅ solid electrolyte	N/A	5.4 mAh cm ⁻² , (0.85 mA cm ⁻²)	41%	200	90.7%
		Li ₃ Bi/LiF artificial layer		6.4 mAh cm ⁻² , (3.40 mA cm ⁻²)	48.4%	200	91.5%*
				6.0 mAh cm ⁻² , (5.1 mA cm ⁻²)	45.4%	200	91.0%*

Abbreviated items:

DND: double-layer nanodiamond; Co/N-PCNS: cobalt -embedded nitrogen -doped porous carbon nanosheets; TCF: tubular carbon fabric.

** The capacity exhibited slight increase in the initial tens of cycles (Fig. R5). Therefore, the capacity retention is calculated based on the ratio of capacity at the last cycle and the maximum capacity achieved over cycling.*

This table could be found in Table S1.

2. The authors mention that their “protected porous Li electrode enables stable operation of a Li-S battery of a high sulfur loading (6.8 mg cm^{-2}) at a high current density (4.0 mA cm^{-2}) for more than 200 cycles” and “At $23 \text{ }^\circ\text{C}$, the discharge capacity achieved at a cathode current density of 4.0 mA cm^{-2} (anode current density 3.4 mA cm^{-2}) was 4.14 mAh cm^{-2} ”. Again, while this result is impressive in the absence of LiNO_3 , 4.14 mAh cm^{-2} , is not going to be a practical areal capacity for the future Li-S battery. The math is simple- the areal capacity of the graphite anode in commercial LIB is $2.5\text{-}3.5 \text{ mAh cm}^{-2}$ and for a Li-S battery with a lower operating voltage ($\approx 2.1 \text{ V}$ compared to ≈ 3.7 of that of LIB) to rival LIB, the areal capacity should be at least 6 mAh cm^{-2} . Given the excess of pretty much every single component in the Li-S battery (for example you are using $100 \text{ }\mu\text{L}$ liquid electrolyte, additional protection layers, a carbon cloth current collector), even at 6 mAh cm^{-2} and gravimetric capacities above $1000 \text{ mAh g}_S^{-1}$, translation to LIB is questionable. I suggest avoid using terms such as high current density when both the areal and gravimetric capacity are not satisfactory compared to the practical levels.

Response: We appreciate the suggestion, and we have revised the discussion using more appropriate terms. We have also assembled full batteries with an even higher sulfur loading (10.2 mg cm^{-2}) and tested the battery at practically high current densities

of 4.0, 6.0 and 8.0 mA cm⁻², with details in the following response.

3. Even though 6.8 mg cm⁻² is certainly a high sulfur loading, a gravimetric capacity of 800 mAh g⁻¹s at such a low current density (1 mAh cm⁻²) means that two critical issues of polysulfides shuttle and Li anode stress are not truly examined in these batteries, the latter in particular. I suggest assembling test cells with high areal capacity cathodes (> 6 mAh cm⁻²) over a few tens of cycles to ensure true examination of the full range of technical challenges that are present in a practical Li-S battery. Perhaps use a higher sulfur-loading cathode and monitor the performance at room temperature while areal capacity is higher than 6 mAh cm⁻². And please plot the Coulombic efficiency data in a more visible fashion, perhaps magnify it in the range of 90 to 100 %.

Response: We agree that a high areal capacity and a high current density would be required to examine all the challenges in practical Li-S batteries. We have now tested an additional battery with a 50% higher sulfur loading (10.2 mg cm⁻²) at even higher current densities, as shown in Fig. R5. Under these conditions, the concentration of Li₂S₈ intermediate can rise to 0.98 M in the cathode region during discharge, whose shuttle effect would lead to significant side reactions if there was no adequate anode protection, and the Li anode stress would seriously deteriorate a planar Li anode as shown previously by Jiao et al.²⁰.

At 45 °C, this battery delivered discharge capacities of 7.53 mAh cm⁻² and 7.05 mAh cm⁻² at 4.0 mA cm⁻² and 6.0 mA cm⁻² respectively. Even at 8.0 mA cm⁻², it could achieve 6.6 mAh cm⁻² (Fig. R5b). When we lowered the temperature to 23 °C as suggested, the discharge capacity decreased likely due to the slower reaction kinetics in the high-loading electrode (Fig. R5a), as previous shown by Xiao et al.²¹. As shown in Fig. R5(c, d), the high-loading battery (10.2 mg cm⁻²) could be cycled at 4.0 and 6.0 mA cm⁻² at 45 °C for more than 200 cycles stably, with an areal capacity constantly

above 7.5 and 6.8 mAh cm⁻², respectively. If cycled at 8.0 mA cm⁻², the battery could last for about 180 cycles. We hope that with this additional set of data, we have shown the feasibility of our design to address the issues in a practical Li-S battery.

As also suggested, we have plotted all the C.E. values within a smaller range to make them more visible as shown in Fig. R5d and Fig. 6(d, e).

Fig. R5 (a, b) Voltage profiles of Li-S batteries assembled with MFC-Li/CNF electrode with a sulfur loading of 10.2 mg cm⁻² (theoretical areal capacity 17.1 mAh cm⁻²) at 23 °C (a); and at 45 °C (b); (c) cyclability at 4.0, 6.0 and 8.0 mA cm⁻² at 45 °C and (d) the corresponding coulombic efficiency. *This figure could be found in Fig. 6 and Fig. S26.*

4. Given that your approach is a two-step process, have you conducted cycling tests with 1) just porous Li electrode, comprising fibrous networks of a Li shell and a carbon core with no coating on the surface, and 2) a lithium foil coated with your proposed

composite layer of Li_3Bi and LiF (if its formation is at all possible on a bare lithium foil). Also, it would be interesting to examine the performance in the presence of LiNO_3 additive in the electrolyte.

Response: We have indeed tested the first type of electrode before as the reviewer suggested, using sulfur cathode with a moderate loading (2.8 mg cm^{-2}) and current (2.0 mA cm^{-2}). As shown in Fig. R6a, a Li-S/CNF electrode, which comprised only the network of Li shells and carbon cores, did not perform well, as lithium polysulfides could corrode the electrode quickly. We have now also tested the second type, a planar Li electrode coated with Li_3Bi (MFC-Li). Although this electrode was much more stable than bare Li, it suffered from a severe capacity decay in the first 100 cycles (Fig. R6a). The results are consistent with our expectation; neither the porous structure nor the coating alone would stabilize the anode in a practical Li-S battery.

Adding LiNO_3 did improve the cycling stability (Fig. R6b), likely because of the healing of SEI by LiNO_3 . However, the stabilizing effect is limited and short-living for both bare Li and Li-S/CNF, likely due to the continuous consumption of LiNO_3 . MFC-Li showed an improved stability with LiNO_3 at 2.0 mA cm^{-2} , but the combination would be less effective when the current density rose to 4.0 mA cm^{-2} . Nonetheless, the performance is all inferior to that of MFC-Li/CNF, confirming the effectiveness of our design strategy.

Fig. R6 (a) Cycling performance of Li-S batteries assembled with Li-S/CNF, MFC-Li

or bare Li anode at 2.0 mA cm^{-2} (based on the area of cathode) with an areal sulfur loading of 2.8 mg cm^{-2} , without the addition of LiNO_3 . (b) Cycling performance achieved with LiNO_3 additive (5 wt% LiNO_3 in 100 μL liquid electrolyte) at 2.0 or 4.0 mA cm^{-2} . *This figure could be found in Fig. S24.*

5. The authors should present discussions and argument on why they think their approach is a better alternative for the simple and commonly used approach of using LiNO_3 as the electrolyte additive. As far as I'm concerned, the literature of Li-S is overwhelmed with similar performance metrics as the authors present in this work where an in-situ SEI layer is formed on the Li metal by using LiNO_3 or other additives in the electrolyte composition.

Response: As shown in the preceding response, the benefit of LiNO_3 is limited and short living, which agrees with previous work showing that native SEI formed from LiNO_3 can sustain for only tens of cycles²². Besides, Jozwiuk et al. showed that the reduction of LiNO_3 additive can trigger the unwanted gas (N_2O and N_2) evolution significantly, leading to the battery package swelling²³. Also, the combination of LiNO_3 , sulfur and carbon forms a hazardous composition similar to the black gun powder²⁴. Considering these potential drawbacks, the LiNO_3 additives might not be relied heavily upon in the future.

We have now included this brief discussion on LiNO_3 at the end of the result section.

In summary, I find the approach interesting and novel. Battery manufacturers are not going to be fond of formation of ex-situ protection layers or composite Li electrode given the complexity of the procedures and the demand for the whole process to be conducted under Ar. Nonetheless, I believe this work would be of the interest to the

scientific community. The results even though not impressive in the general literature of Li-S can be considered as very good given the absence of LiNO₃. To ensure the true suitability of this Li coating, test cells must be configured with high areal capacity cathodes to allow for passage of realistic current densities.

Response: We do agree with the reviewer that it would be important to look at the scalability of the method. As indicated, the porous Li electrode (Li/S-CNF) needs to be fabricated in the inertial Ar gas atmosphere to prevent side reactions between Li and N₂/O₂/CO₂. Inspired by the reviewer, we further studied the dry air stability of the precursor and the as-fabricated Li electrode to see whether they are stable during the protective coating and the battery assembly process. We show that BiF₃-P₂S₅ in DME precursor remained stable after the dry air exposure as shown in the UV-Vis spectra (Fig.R7 (a, b)). Meanwhile, for the MFC-Li/CNF, the electrode components (LiF, Li₃Bi and Li₂S-P₂S₅ solid electrolyte) are stable compounds, and the Li₃Bi/LiF coating could to some extent prevent the side reactions between Li and N₂/O₂/CO₂, which lowers the Li loss from 46% in the case of air-exposed Li-S/CNF to 7.4% (Fig. R7(c, d)).

Fig. R7 Dry air stability test. (a) UV-Vis spectra of the supernatant solution (diluted to

25% of its initial concentration) of BiF₃-P₂S₅ before and after dry air exposure. (b) Photography of the BiF₃-P₂S₅ in DME (50 mM BiF₃) in the dry air environment. (c) Li stripping (0.5 mA cm⁻²) curves of Li-S/CNF and MFC-Li/CNF before and after dry air exposure. (d) Photography of the electrodes in the dry air environment. *This figure could be found in Fig. S30.*

Anhydrous CaCl₂ (200 mg) was put in the vessel (20 mL) to create dry air environment. The electrodes (8 mm in diameter) for testing were punched from the same Li/S-CNF or MFC-Li/CNF pellet (18 mm), to guarantee that their properties were almost identical. The nitrogen and oxygen contained in the vessel are excessive, which can fully react ~34 mg Li to form Li₃N and Li₂O.

We are beyond grateful for the reviewers' comments, which are really helpful for improving the quality of this paper.

References

1. Peng, Z. *et al.* Stabilizing Li/electrolyte interface with a transplantable protective layer based on nanoscale LiF domains. *Nano Energy* **39**, 662–672 (2017).
2. Zhao, J. *et al.* Surface fluorination of reactive battery anode materials for enhanced stability. *J. Am. Chem. Soc.* **139**, 11550–11558 (2017).
3. Lin, D. *et al.* Conformal lithium fluoride protection layer on three-dimensional lithium by nonhazardous gaseous reagent freon. *Nano Lett.* **17**, 3731–3737 (2017).
4. Pang, Q., Liang, X., Kochetkov, I. R., Hartmann, P. & Nazar, L. F. Stabilizing Lithium Plating by a Biphasic Surface Layer Formed In Situ. *Angew. Chemie Int. Ed.* **57**, 9795–9798 (2018).
5. Yan, C. *et al.* An Armored Mixed Conductor Interphase on a Dendrite-Free

- Lithium-Metal Anode. *Adv. Mater.* **30**, 1804461 (2018).
6. Lang, J. *et al.* One-pot solution coating of high quality LiF layer to stabilize Li metal anode. *Energy Storage Mater.* **16**, 85–90 (2019).
 7. Sun, K. *et al.* Interaction of CuS and sulfur in Li-S battery system. *J. Electrochem. Soc.* **162**, A2834–A2839 (2015).
 8. Chen, Y., Freunberger, S. A., Peng, Z., Bardé, F. & Bruce, P. G. Li–O₂ battery with a dimethylformamide electrolyte. *J. Am. Chem. Soc.* **134**, 7952–7957 (2012).
 9. Chu, H. *et al.* Achieving three-dimensional lithium sulfide growth in lithium-sulfur batteries using high-donor-number anions. *Nat. Commun.* **10**, 188 (2019).
 10. Pang, Q., Liang, X., Shyamsunder, A. & Nazar, L. F. An In Vivo Formed Solid Electrolyte Surface Layer Enables Stable Plating of Li Metal. *Joule* **1**, 871–886 (2017).
 11. Zhang, Q. *et al.* Synergetic effects of inorganic components in solid electrolyte interphase on high cycle efficiency of lithium ion batteries. *Nano Lett.* **16**, 2011–2016 (2016).
 12. Liu, Y. *et al.* An Ultrastrong Double-Layer Nanodiamond Interface for Stable Lithium Metal Anodes. *Joule* (2018).
 13. Cha, E. *et al.* 2D MoS₂ as an efficient protective layer for lithium metal anodes in high-performance Li–S batteries. *Nat. Nanotechnol.* **13**, 337 (2018).
 14. Nanda, S., Gupta, A. & Manthiram, A. A Lithium–Sulfur Cell Based on Reversible Lithium Deposition from a Li₂S Cathode Host onto a Hostless-Anode Substrate. *Adv. Energy Mater.* 1801556 (2018).
 15. Chen, L. *et al.* Directly formed alucone on lithium metal for high-performance Li batteries and Li–S batteries with high sulfur mass loading. *ACS Appl. Mater. Interfaces* **10**, 7043–7051 (2018).

16. Tang, W. *et al.* Lithium Silicide Surface Enrichment: A Solution to Lithium Metal Battery. *Adv. Mater.* **30**, 1801745 (2018).
17. Liu, S. *et al.* Superhierarchical Cobalt-Embedded Nitrogen-Doped Porous Carbon Nanosheets as Two-in-One Hosts for High-Performance Lithium–Sulfur Batteries. *Adv. Mater.* **30**, 1706895 (2018).
18. Cai, W. *et al.* The Dual-Play of 3D Conductive Scaffold Embedded with Co, N Codoped Hollow Polyhedra toward High-Performance Li–S Full Cell. *Adv. Energy Mater.* **8**, 1802561 (2018).
19. Chang, J. *et al.* Flexible and stable high-energy lithium-sulfur full batteries with only 100% oversized lithium. *Nat. Commun.* **9**, 4480 (2018).
20. Jiao, S. *et al.* Behavior of lithium metal anodes under various capacity utilization and high current density in lithium metal batteries. *Joule* **2**, 110–124 (2018).
21. Xiao, J. Understanding the lithium sulfur battery system at relevant scales. *Adv. Energy Mater.* **5**, 1501102 (2015).
22. Adams, B. D. *et al.* Long term stability of Li-S batteries using high concentration lithium nitrate electrolytes. *Nano Energy* **40**, 607–617 (2017).
23. Jozwiuk, A. *et al.* The critical role of lithium nitrate in the gas evolution of lithium–sulfur batteries. *Energy Environ. Sci.* **9**, 2603–2608 (2016).
24. Qu, C. *et al.* LiNO₃-free electrolyte for Li-S battery: A solvent of choice with low K_{sp} of polysulfide and low dendrite of lithium. *Nano Energy* **39**, 262–272 (2017).

REVIEWERS' COMMENTS:

Reviewer #1 (Remarks to the Author):

The reviewer is satisfied with the manuscript: "Rational design of spontaneous to form a protected porous Li electrode for a stable Li-S battery" after the revision was made by the authors: Zhao et al. The current version is appropriate for publication in Nat. Comm.

Reviewer #3 (Remarks to the Author):

The authors have handled the comments/criticisms very well and the manuscript has been suitably revised. I strongly believe this fine paper is ready for publication in Nature Communication in its current form and without any further delay. It is certainly of great interest to the lithium-metal based batteries community, and the findings are novel and solidly supported. I particularly appreciate the revised presentation of CE data in Fig. 6 d and e, which highlights the great results you achieved in the absence of LiNO₃.

Mahdokht Shaibani

Response to the reviewers' comments

Reviewer #1 (Remarks to the Author):

The reviewer is satisfied with the manuscript: "Rational design of spontaneous to form a protected porous Li electrode for a stable Li-S battery" after the revision was made by the authors: Zhao et al. The current version is appropriate for publication in Nat. Comm.

Response: We are beyond grateful to hear the reviewer's positive comments on our manuscript.

Reviewer #3 (Remarks to the Author):

The authors have handled the comments/criticisms very well and the manuscript has been suitably revised. I strongly believe this fine paper is ready for publication in Nature Communication in its current form and without any further delay. It is certainly of great interest to the lithium-metal based batteries community, and the findings are novel and solidly supported. I particularly appreciate the revised presentation of CE data in Fig. 6 d and e, which highlights the great results you achieved in the absence of LiNO_3 .

Mahdokht Shaibani

Response: We thank the reviewer sincerely for raising the constructive comments.